# Biodegradation in Freshwater: Comparison Between Compostable Plastics and Their Biopolymer Matrices

**DOI:** 10.3390/polym17162236

**Published:** 2025-08-17

**Authors:** Valerio Bocci, Martina De Vivo, Sara Alfano, Simona Rossetti, Francesca Di Pippo, Loris Pietrelli, Andrea Martinelli

**Affiliations:** 1Water Research Institute, CNR-IRSA, National Research Council, Monterotondo, 00015 Rome, Italy; valerio.bocci@irsa.cnr.it (V.B.); simona.rossetti@irsa.cnr.it (S.R.); francesca.dipippo@irsa.cnr.it (F.D.P.); 2PhD Program in Evolutionary Biology and Ecology, Department of Biology, University of Rome ‘Tor Vergata’, 00133 Rome, Italy; 3Department of Chemistry, Sapienza University of Rome, 00185 Rome, Italy; devivo.1904692@studenti.uniroma1.it (M.D.V.); sara.alfano@uniroma1.it (S.A.); 4Legambiente, Scientific Committee, 00199 Rome, Italy

**Keywords:** biopolymer, plastisphere, polylactic acid (PLA), Mater-Bi (Mb), poly(3-hydroxybutyrate-co-3-hydroxyvalerate) (PHBV), poly(butylene adipate-co-terephthalate) (PBAT), compostable, shopping bag, disposable dish

## Abstract

Plastic pollution in freshwater ecosystems is an increasing environmental concern, prompting the search for biodegradable polymer (BP) alternatives. However, their degradation in natural aquatic environments remains poorly investigated and understood. This four-month in situ study compared the degradation in a lentic freshwater ecosystem of two compostable items, Mater-Bi^®^ shopping bag and disposable dish, with their respective pure polymer matrices, poly(butylene adipate-co-terephthalate) (PBAT) and polylactic acid (PLA). Additionally, biodegradable poly(3-hydroxybutyrate-co-3-hydroxyvalerate) (PHBV) and oil-based polypropylene (PP) were also tested. Changes in morphology, chemical composition and thermal and mechanical properties, as well as microbial colonization, were analyzed over time. A validated cleaning protocol was employed to ensure accurate surface analysis. Results showed detectable but limited degradation of pure polymers and their matrices in commercial products after 120 days of immersion with variations observed among polymer materials. Compostable materials exhibited significant leaching of fillers (starch, inorganic particles), leading to morphological changes and fragmentation. PHBV showed the fastest degradation among tested polyesters. PP exhibited only minor surface changes. Microbial colonization varied with polymer structure and degradability, but long-term degradation was limited by polymer properties and the gradual development of the plastisphere. This study highlights that standard laboratory tests may overestimate the environmental degradability of BPs and emphasizes the importance of in situ assessments, careful cleaning procedures and property characterizations to accurately assess polymer degradation in freshwater systems.

## 1. Introduction

Plastic pollution in freshwater ecosystems has emerged as a critical global concern due to its far-reaching ecological consequences. Rivers, lakes and wetlands are especially vulnerable to the accumulation of plastic debris, owing to their durability and persistence. Plastics pose significant environmental threats, including physical harm to aquatic organisms, disruption of food webs and the release of toxic contaminants into water systems, thereby exacerbating the degradation of freshwater habitats [1]. These impacts have been well documented in marine environments; however, increasing evidence suggests that similar effects are occurring in freshwater ecosystems, where plastic accumulation can modify habitat structure, degrade water quality and impact organisms across multiple trophic levels [2,3,4].

Plastics, composed of a broad range of synthetic polymers, are used extensively across sectors such as packaging, automotive manufacturing and construction. However, the greatest environmental burden arises from the excessive production and use of short-life, low-value disposable products, such as single use plastics (SUPs). These items often result in complex waste streams that are costly and difficult to manage, often leading to widespread environmental dispersion.

In response to the growing plastic pollution crisis, increasing attention has been directed towards the development of sustainable alternatives. Biodegradable polymer (BP) materials, derived from renewable biological sources, have been proposed as a potential solution. These materials can degrade into carbon dioxide, water and inorganic compounds through the action of microorganisms and abiotic environmental factors [5]. BPs, including polylactic acid (PLA), polyhydroxyalkanoates (PHA), and starch-based blends, are considered potentially faster-degrading alternatives to conventional plastics, especially within the context of regulatory frameworks like the European Union’s Single-Use Plastics Directive (EU/2019/904) [6], which aims to reduce plastic waste.

However, the degradation of these BPs under natural environmental conditions, particularly in freshwater ecosystems, is little studied given the multitude of parameters to be controlled [7]. Standardized laboratory tests frequently fail to replicate the complexity of natural ecosystems, where microbial community composition, temperature and water chemistry vary significantly [8,9]. Consequently, the degradation rates observed under controlled conditions may not accurately reflect those in open environments. Furthermore, once plastic items are transported through the aquatic ecosystems, abiotic factors such as temperature fluctuations and UV radiation can alter their mechanical and physico-chemical properties. Additionally, biofilm microorganisms colonizing plastic surfaces, collectively known as the plastisphere [10], can further influence degradation by secreting extracellular enzymes, degrading additives and initiating bio-fragmentation [11].

Biodegradation typically proceeds through multiple stages: initial enzymatic bio-fragmentation, followed by the assimilation and mineralization of resulting oligomers and monomers, although under environmental conditions this process may be influenced by both biotic and abiotic factors and may result in incomplete degradation, potentially leading to persistent microplastic residues [12]. Moreover, under environmental conditions, the biodegradation process can be slow [8] and the plastisphere, composed of diverse microbial taxa, may influence degradation dynamics. These plastic-attached communities have indeed been shown to promote changes in surface material properties, including chemical composition, morphology and weight loss [13,14].

Recognizing the limitations of laboratory testing, scientific advisory bodies such as SAPEA (2020) [15] have stressed the importance of in situ assessments of BP degradation in open environments, including freshwater ecosystems. Field-based experiments are essential to better understand the fate of biodegradable materials and to evaluate how biotic and abiotic interactions influence their degradation over time. The complexity of this issue is increased by the lack of standardized analytical methods for assessing polymer modifications in natural environments, including consistent protocols for surface cleaning. These gaps hinder the reproducibility of results and limit meaningful comparisons across studies.

A four-month field experiment was conducted in a lentic ecosystem to provide critical insights into the degradation of BPs (defined here as materials containing a high polymer as an essential component, according to ISO 472 or ASTM 883) in natural aquatic environments. This timeframe was chosen to evaluate the degradation of biopolymers marketed as rapidly degradable, acknowledging that while non-biodegradable polymers degrade slowly, data on biopolymer degradation in freshwater environments remain limited.

This study investigates the complex interplay between polymer physicochemical properties and biofilm development under natural freshwater environments, by analysing potential changes in material morphology, chemical composition, thermal and mechanical properties and microbial colonization over time.

Considering that many common compostable products are composite materials containing fillers and additives that may influence biotic and abiotic degradation processes, we selected both commercial items and their corresponding pure polymer matrices. Specifically, compostable disposable dishes based on polylactic acid (PLA) and pure PLA films, compostable shopping bags composed of Mater-Bi^®^ (a composite of starch and poly(butylene adipate-co-terephthalate) (PBAT) and pure PBAT films, biodegradable poly(3-hydroxybutyrate-co-3-hydroxyvalerate) (PHBV) and polypropylene films (used as a rapid biodegradable and non-biodegradable petroleum-based reference polymers, respectively) were utilized. To ensure the accuracy of our assessments, a cleaning protocol was developed, validated and applied to effectively remove biofilm and chemical residuals from sample surfaces prior to characterization.

## 2. Materials and Methods

### 2.1. Materials

In the present research, two biodegradable daily-life plastic products were selected, and their corresponding virgin polymers, which served as their primary constituents, were included in the study for comparative purposes. Rapidly biodegradable and non-degradable polymers were also analysed. In Table 1, the list of the investigated samples, sample codes, main characteristics, thickness and supplier are reported.

Virgin PLA, PBAT and PHBV samples were obtained in the form of films by compressing about 600 mg of polymer pellets (PLA-r, PBAT) or powder (PHBV) between two poly(tetrafluoroethylene) foils within a 120 μm thick aluminium frame with a 6 × 6 cm^2^ window. The assembly was heated at 170 °C (PLA-r and PHBV) or 130 °C (PBAT) for 1 min and pressed at 400 kPa for 1 min. The obtained film was cooled naturally at room temperature and characterized for its pristine properties after a week of stabilization.

### 2.2. Experimental Design and Field Activity

The experiment was conducted at a selected coastal site of Lake Bracciano (42°06′19.0″ N 12°11′23.0″ E, 164 m a.s.l., Latium, Italy), a crucial drinking water reservoir in Central Italy. For each material, multiple specimens required for all analyses were attached to a support and enclosed within a rigid cage. It consisted of an iron frame with a movable top to facilitate easy sample access (Appendix A). The cage was then secured and submerged approximately 20 cm below the water surface with the mesh, allowing for unimpeded water exchange driven by natural waves and currents. Prior to deployment, the samples were rinsed with a 70% ethanol (Darmstadt, Germany) solution for preliminary surface cleaning and sterilization [16].

Beginning in May 2024, three specimens for each analysis were retrieved at 7, 14, 30, 60, 90 and 120 days for surface characterization and at 30, 60, 90 and 120 days for bulk characterization. Simultaneously, the water parameters were recorded (Appendix A). All material samples were cleaned as detailed in Section 2.3 before analysis. Additionally, two material coupons were gently washed with a sterile saline solution (0.9%) to remove non-attached organisms, fixed in 2.5% formaldehyde and 96% ethanol and stored at −20 °C until catalysed reporter deposition fluorescence in situ hybridisation (CARD-FISH) analyses and confocal laser scanning microscopy (CLSM) (Olympus Corp., Tokyo, Japan) observations. Three replicates were washed with sterile saline solution (0.9%) and stored at −20 °C for subsequent DNA extractions [17] and qPCR analyses. The experimental set up is shown in Appendix A.

### 2.3. Sample Cleaning Procedures

Considering that all surface analyses, including SEM, ATR-FTIR and contact angle measurements, are significantly influenced by the presence of any surface contaminants and it is not uncommon in the literature to find misinterpretations of degradation processes often caused by uneven sample cleaning, a multiple stages cleaning procedure was performed. Three different procedures to remove biofilm microorganisms and chemical residuals from sample surfaces prior to characterization were tested. Firstly, the protocol of the DNeasy PowerSoil Pro Kit (QIAGEN, Germantown, MD, USA) currently used for DNA extraction was applied [18] (Method 1). The second method (Method 2), a modification of the protocol by Devi et al. [19], consisted of the following steps: (i) immersion of samples in a 2% *w*/*v* sodium dodecyl sulphate (SDS) solution for 3 h; (ii) sonication for 1 h; (iii) rinsing of samples with distilled water; (iv) rinsing of samples with 70% *v*/*v* ethanol; (v) drying of samples at room temperature; (vi) removal of residual moisture using a rotary evaporator. For the PLA-d-X and PLA-r-X films, step 4 was not used to avoid change in sample crystallinity favoured by EtOH [20]. The third method (Method 3) comprised the sequential application of the first and second method.

Method 3 was also tested on pristine plastic items to exclude any modifications to their surface chemical, physical or morphological properties, prior to ATR-FTIR and SEM analyses as well as contact angle measurements.

### 2.4. Spectroscopic Characterization

Infrared spectra of the sample surface were acquired by a Nicolet 6700 FTIR (Waltham, MA, USA) equipped with a Specac Golden Gate attenuated total reflection (ATR) (Orpington, UK device endowed with a single reflection diamond element. The spectra were recorded in a 4000–600 cm^−1^ spectral range by co-adding 100 scans per spectrum at a resolution of 4 cm^−1^. To ensure comparability, all reported spectra and the results obtained from their analysis were normalized by selecting adequate absorption band intensities, specific for each material, as illustrated in the spectroscopic results and discussion.

### 2.5. Water Contact Angle Measurements

Static water contact angle (WCA) measurements were accomplished using a custom-made apparatus. At least five 5 μL water drops per sample were deposited on the plastic surface, and their images were recorded using a digital camera. WCA was determined by the ImageJ (Image Processing and Analysis in Java, v.1.54b) dedicated plug-in. The reported WCAs are the average value (±standard deviations) of measurements carried out on at least five drops deposited on different sample zones.

### 2.6. Thermal Characterization

The sample thermal behaviour was investigated by differential scanning calorimetry (DSC) and thermogravimetric analysis (TGA).

A Mettler Toledo DSC 822e instrument (Columbus, OH, USA) was used to obtain DSC thermograms by heating about 4–6 mg of sample from 35 °C to 200 °C under N2 flow (30 mL min^−1^).

Thermogravimetric analysis was performed using a Mettler TG50-MT5 (Columbus, OH, USA) with a heating rate of 10 °C min^−1^ from 25 °C to 500 °C under N2 flow (30 mL min^−1^).

### 2.7. Morphological Characterization

The morphology of the sample surface was investigated by scanning microscopy using a field emission scanning electron microscope (AURIGA, Zeiss, Jena, Germany). The samples were chromium-sputtered before the analysis.

### 2.8. Characterization of Mechanical Properties

Tensile stress–strain tests on virgin and immersed samples at different time points were carried out by a universal testing machine (Instron 4502, High Wycombe, UK) at room temperature, using a 2 kN load cell at a crosshead speed of 5 mm min^−1^. The dumbbell specimens were die-cut from commercial plastics or from polymer films with shape and dimensions conforming to ASTM D638, Type IV.

The Young modulus (E), tensile strength (TS) and elongation at break (ε_b_) were determined from stress–strain curves obtained by reporting the apparent stress σ = F/A (MPa), where F is the tensile force and A is the initial cross-sectional area of each test specimen, versus the strain ε = (L_0_ − L)/L_0_, where L_0_ and L are the initial and the deformed sample length, respectively. In particular, the Young modulus was calculated from the slope of the initial linear zone in the stress–strain curve. The results are reported as average values (±standard deviations) on at least three experiments conducted on different specimens.

The mechanical properties were evaluated for up to 90 days for PLA-r, PLA-d and PBAT, and for up to 60 days for Mb, due to the onset of fissuring and cracking in the specimens beyond these timeframes.

### 2.9. Catalysed Reporter Deposition Fluorescence in Situ Hybridisation (CARD-FISH) and Confocal Laser Scanning Microscopy (CLSM)

CARD-FISH was performed using horseradish peroxidase-labelled oligonucleotide probes (Biomers, Ulm, Germany) and signal amplification with fluorescein-labelled tyramides. EUB338mix probes and DAPI staining (1.5 µg mL^−1^, Vector Labs, Milan, Italy) were used to identify bacteria and total microorganisms in the biofilm, respectively. Photosynthetic prokaryotic and eukaryotic cells were visualized based on their auto-fluorescent pigments.

Biofilm samples were observed using a CLSM FV1000 (Olympus Corp., Tokyo, Japan) in multichannel mode to visualize microbial cell clusters on all materials at various sampling times (representing different biofilm development stages). Three-dimensional (3D) images were reconstructed from two-dimensional (2D) cross-sectional images (x–y plane; 0.5 µm intervals), and 3D volume renderings in blend mode were generated using Imaris 6.2.0 software (Bitplane AG, Zurich, Switzerland).

### 2.10. DNA Extraction from Surface-Attached Communities

DNA extraction from materials was performed using the DNeasy PowerSoil Pro Kit (QIAGEN, Germantown, MD, USA), following the manufacturer’s instructions. The purified DNA from each sample was eluted in 50 µL of sterile Milli-Q water and stored at −20 °C. The success of the extraction and DNA quantification were assessed using a Nanodrop 3300 spectrophotometer (Thermo Scientific, Monza, Italy).

### 2.11. Quantitative PCR (qPCR)

Quantitative PCR (qPCR) was employed to quantify the total microbial load by targeting the 16S rRNA gene in DNA extracted from all the material experimental replicates [21]. The qPCR was performed using a CFX96™ Real-Time PCR Detection System (Bio-Rad Laboratories, Hercules, CA, USA) in 96-well plates. Each 20 µL reaction mixture contained 2 µL of DNA template (3–5 ng µL^−1^), 10 µL of SsoAdvanced Universal SYBR Green Supermix (Bio-Rad), 1 µL of primers (10 µM) and 6 µL of sterile Milli-Q water. The 16S rRNA gene amplification program consisted of maintaining the sample at 95 °C for 2 min, followed by 35 cycles of 95 °C for 15 s, 60 °C for 30 s and 72 °C for 15 s. A melt curve analysis was performed from 60 °C to 95 °C with increments of 0.5 °C every 5 s. All reactions were conducted in triplicate for each DNA sample. Data were analysed using CFX Manager™ software (version 3.1, Bio-Rad, Italy). The quantity of target genes in unknown samples was determined using a standard curve (Ct value versus log of initial gene copy number) generated from serial dilutions of a positive control. The 16S rRNA gene used as a positive control was amplified by PCR and quantified using a NanoDrop spectrophotometer, and the gene copy number per µL of genomic DNA solution was calculated as described by Czekalski et al. [22]. Results were reported as the average of measurements with standard deviations, and the amount of 16S rDNA was normalized to the plastic surface area (gene copy number per cm^−2^).

## 3. Results

### 3.1. Cleaning Procedure Test

Surface analysis techniques such as SEM, ATR-FTIR and contact angle measurements are highly sensitive to surface contamination. Although various cleaning protocols have been proposed, no standardized method currently ensures the complete removal of biological material, such as plastisphere microorganisms, from plastics exposed to natural environments. An effective cleaning procedure should be applicable across diverse polymer types and independent of their chemical composition. For example, while alkaline solutions efficiently remove organic residues from polyethylene, they may degrade polyesters via hydrolysis. To ensure reliable data, cleaning methods must be validated on virgin polymer samples to confirm they do not induce morphological or compositional changes beyond biofilm removal. The procedure should not leave chemical residues (e.g., surfactants), and mechanical treatments must preserve the surface morphology. For multi-step protocols, the effect of each stage should be evaluated separately using control samples. These criteria are essential to avoid analytical artefacts and ensure comparability across studies. A previous study demonstrated that a DNA extraction protocol effectively removed microorganisms from Mb and PLA films without altering surface properties [17]. In the present work, two cleaning methods (Method 1 and Method 2) were tested on PBAT-30 films. SEM images (Appendix A) revealed that neither method alone removed all organic matter. However, their sequential application (Method 1 followed by Method 2) successfully eliminated surface residues, as shown in Appendix A. This combined protocol was therefore adopted for all samples. Its effectiveness was confirmed by ATR-FTIR analysis of PBAT-30 films before and after cleaning (Appendix A). The spectrum of the cleaned sample matched that of pristine PBAT-30, indicating complete removal of organic contaminants. In contrast, uncleaned samples showed additional bands in the 3700–3000 cm^−1^, 1700–1600 cm^−1^ and 900–1200 cm^−1^ regions, consistent with polysaccharides and proteins. To confirm that the cleaning process did not damage the samples, we subjected all of them to the selected cleaning procedure (Method 3) before immersion in freshwater. Samples were characterized using ATR-FTIR, WCA and SEM, and the results showed that the surface properties were unchanged compared to the virgin samples. Consequently, we adopted this new sequential cleaning procedure for all samples.

### 3.2. Sample Characterization

In this section, the variations in chemical, physical, morphological and mechanical properties of the samples were analysed as a function of immersion time. Analyses involved a parallel comparison of the surface and bulk features of plastic items with that of their respective matrix polymers. At the end of these analyses, a discussion on the results is given.

#### 3.2.1. PLA-d and PLA-r Surface Characterization

Like many commercial plastic items, compostable dishes typically contain fillers and additives, added to the polymer matrix. Appendix A displays the spectra of PLA-r and PLA-d, along with the spectra of two identified filler materials: talc (characteristic sharp band at 665 cm^−1^) and calcium carbonate (CaCO_3_, with a large absorption at 1392 cm^−1^ and sharp bands at 711 cm^−1^ and 867 cm^−1^). Moreover, the PLA-d-0 spectrum, exhibiting the strong C=O stretching absorption band at 1749 cm^−1^, displays an additional carbonyl band at 1718 cm^−1^ assigned to poly(butylene succinate) (PBS), as confirmed by the DSC analysis detailed later. The fillers were added to impart whiteness, enhance mechanical properties, reduce material costs and act as a nucleating agent, and PBS was added to increase the material toughness [23]. The evolution of the spectra of PLA-d samples as a result of immersion in freshwater revealed a progressive intensity reduction of the band due to the additives (Appendix A). The intensities of these bands were normalized with respect to the intensity of the PLA C=O stretching band at 1749 cm^−1^, not superimposed to those of the fillers, and plotted as a function of immersion time in Figure 1a.

The variation of the normalized absorption of the bands at 665 cm^−1^ and 1386 cm^−1^, shown in Figure 1a, indicates that the filler content decreases up to 14 days, remaining nearly constant for longer immersion times. The fact that the filler bands were clearly detectable in all the explored period and considering that ATR-FTIR analysis has a penetration depth in the order of one micron or less imply that the sample loses the fillers within this outer thin region. In contrast, the decrease in the band intensity at 1718 cm^−1^ was very rapid, levelling off after 7 days.

Given that the significant contribution of the additive to PLA-d spectra and the effect of its concentration changes could mask potential polymer matrix degradation signs, ATR-TIR analysis of PLA-r was also conducted over immersion time. Appendix A presents the spectra of selected samples (0, 30 and 120 days). No new bands appeared in the spectra of PLA-r, but there were minor variations in the intensity and shape of certain absorption bands, not attributed to matrix compositional changes but primarily influenced by changes in polymer crystallinity. For example, the intensity of the bands at 1267 cm^−1^, associated with the amorphous phase (marked by arrows in Appendix A) was normalized with respect to the intensity of the C–H bending band at 1452 cm^−1^, which is not significantly affected by crystallinity, and reported as a function of immersion time in Figure 1b. The observed variation of the absorbance ratio indicates a rapid crystallization of PLA promoted by the plasticizing effect of water, already reported in the literature [24]. Although possible hydrolysis reactions were considered, they were not detected by ATR-FTIR analysis. This is likely due to the low concentration of –COOH and –OH groups on the polymer surface caused by factors such as remotion of low-molecular-weight hydrolysis products during the cleaning protocol or their solubilization in water or bio-assimilation, preventing their accumulation on the surface.

SEM analysis shows that PLA film obtained by thermo-pressing polymer pellets (PLA-r-0) is amorphous and exhibits a wavy, uneven surface on a macroscopic scale but appears quite smooth at the sub-microscopic level (Figure 2a). Upon immersion, a typical morphology of twisted and intertwined lamellae appeared on the sample surface, likely resulting from a randomly nucleated cold-crystallization process (PLA-r-30, Figure 2b). At longer immersion times, the lamellar structure became more defined and the surface roughness increased (Figure 2c) because of the progressive hydrolysis of the outermost amorphous fraction, which is more susceptible to chain scission. The effect of immersion is more pronounced on the compostable dish. The initial morphology of the PLA-d-0 is very similar to that of crystalline PLA, exhibiting a crystalline lamellar morphology, in addition to humps likely due to the presence of inorganic fillers (Figure 2d). However, after only 30 days of immersion, significant morphological changes occur, including a roughness increase, delamination, pitting and the formation of large holes (Figure 2e) whose number and dimensions progressively increase over time (Figure 2d). The observed loss of compactness of the sample surface allowed the release of the filler, as already observed from FTIR analysis.

To explore the property variations of the outermost surface, water contact angle measurements were carried out on pristine and immersed samples (Figure 3). Due to the hydrophobicity of PLA, the PLA-r-0 sample exhibits a high WCA of 86°, which rapidly decreases upon immersion, reaching approximately 70–66° after one month. This increase in wettability is attributed to the combined effect of increased surface roughness and the formation of polar groups resulting from the hydrolysis of the outermost surface. The pristine PLA-d sample exhibits higher initial wettability due to the presence of hydrophilic CaCO_3_. Subsequently, the decrease in WCA of the PLA-d sample is slower than that of PLA-r, likely attributed to its higher crystallinity and, consequently, higher resistance to hydrolysis.

However, the sample gradually reaches WCAs lower than those of PLA-r, due to the exposure of the inner filler caused by surface erosion as well as to an increase in surface roughness and the formation of holes.

#### 3.2.2. PLA-d and PLA-r Bulk Characterization

To assess the total amount of filler in the compostable dish and its reduction during immersion in freshwater, the samples were subjected to TGA analysis (Figure 4a).

The pristine PLA-d sample exhibits two distinct weight loss stages. The first one begins at approximately 270 °C, with a maximum decomposition rate at T_d1_^max^ = 340 °C. The second weight-loss event was characterized by a maximum decomposition temperature T_d2_^max^ = 390 °C, a value consistent with that reported for PBS [25]. Upon immersion in water, T_d1_^max^ progressively decreased, leading to a broadening of the decomposition temperature range, primarily due to a reduction in the onset (Figure 4b). Concurrently, the immersion resulted in a reduction in the second weight loss event due to PBS decomposition, evaluated at 380 °C in Figure 4c. The degradation of the more thermally stable PBS component appeared to influence the stability of the PLA fraction, which subsequently decomposed at lower temperature [25]. At the highest temperatures, the residual mass, primarily representing the inorganic filler content and evaluated at 595 °C, decreased with increasing immersion time (Figure 4c).

Comparison of FTIR results (Figure 1a), which evidenced a rapid and large decrease in filler concentration, with those of TGA (Figure 4c), which shows a relatively small overall weight loss of approximately 2%, revealed that the additive leaching primarily affected the outer layer of the sample.

DSC experiments were carried out to investigate the evolution of the thermal behaviour of the samples during immersion in freshwater. Appendix A presents the thermograms of three PLA-r selected samples, exhibiting the typical transitions of predominantly amorphous PLA. At about 60 °C, the glass transition occurs, superimposed by an overshoot peak of PLA-r-60 and PLA-r-120 samples, due to the recovery of enthalpy relaxation that occurred during immersion. At higher temperatures, the exothermic peaks are due to cold crystallization, followed by an endothermic process corresponding to the melting. Prior to melting, an exothermic peak associated with the typical crystal transformation from the α’ form to the more stable α form is evident. The variation of the enthalpy of cold crystallization (ΔH_cc_^PLA^) and melting (ΔH_m_^PLA^) provides key insights into the sample transformations induced by immersion in freshwater (Figure 4d). A simultaneous decrease in ΔH_cc_^PLA^ and increase in ΔH_m_^PLA^A indicates that the initially almost amorphous sample undergoes progressive crystallization. The transformation is likely facilitated by the uptake of small amounts of water, which enhances macromolecular segmental mobility [24]. Appendix A presents the DSC profiles of selected PLA-d samples. The recorded heat flux was normalized with respect to the polymer matrix weight, as determined from TGA results. In addition to the aforementioned PLA features, the thermograms exhibit a further endothermic transition at approximately 113 °C, attributed to the melting of PBS [26]. The DSC thermograms were analysed by evaluating the enthalpy associated with the transitions: PLA cold crystallization (ΔH_cc_^PLA^), PBS melting (ΔH_m_^PBS^) and PLA melting (ΔH_m_^PLA^) (Figure 4e). The results indicate that the pristine PLA-d-0 sample was initially semi-crystalline since ΔH_cc_^PLA^ was lower than ΔH_m_^PLA^. However, the crystallinity that the PLA-d-X samples can reach is lower than that of PLA-r-X as their ΔH_m_^PLA^ values are two to three times lower. Unlike the observations for PLA-r, the decrease in ΔH_cc_^PLA^ was not accompanied by a corresponding increase in ΔH_m_^PLA^, suggesting a progressive reduction in sample crystallinity. Moreover, the decrease in ΔH_m_^PBS^ indicates that PLA-d samples lose the PBS component over time, corroborating FTIR and TGA results (Figure 1a and Figure 4c).

The changes in surface and bulk properties experienced by the samples during immersion did not significantly impact their mechanical properties. Appendix A present representative stress–strain curves for PLA-r and PLA-d samples, respectively. The mechanical properties derived from stress–strain experiments of PLA-r and PLA-d samples, namely modulus (E), tensile strength (TS) and elongation to break (ε_b_), are reported in Figure 4f and Figure 4g, respectively. Figure 4f illustrates the effect of PBS addition in enhancing the toughness of PLA, which is otherwise brittle (Figure 4g) [23]. PLA-d-X, in fact, exhibits yielding and a higher elongation at break than that of PLA-r-X samples. The progressive decrease of PBS content, observed in ATR-FTIR and thermal analysis, led to an increase in the PLA-d modulus and a decrease in tensile strength and elongation at break (Figure 4f). Conversely, PLA-r samples did not display significant variations in mechanical properties as a function of immersion time (Figure 4g). Notably, the error associated with ε_b_ is high due to the presence or development during the immersion of flaws and cracks, which act as precursors to sample fracture in brittle materials. For this reason, it was not possible to test samples immersed for longer than 90 days, as those specimens underwent rupture and fragmentation.

The above reported results indicate that polylactic acid films (PLA-r) retained general stability in freshwater, especially when in a semi-crystalline state. The well-known slow crystallization rate of PLA resulted in a largely amorphous structure of newly prepared film. However, its glass transition temperature (T_g_ = 60 °C), above typical freshwater temperatures, represents a factor that hinders enzymatic degradation [17]. Water immersion facilitates partial swelling, promoting crystallization that further enhances resistance to both abiotic and biotic hydrolysis [27]. The observed change in PLA properties, mainly on the film surface, is consistent with existing literature documenting the slow degradation of PLA in various controlled and uncontrolled environments [28,29,30]. Specifically, studies on compostable plastic carrier bags containing at least 30% PLA in river streams and riparian areas have shown a limited degradation of no more than 5 wt.% within the initial 14 days of a 77-day total period [31]. Furthermore, the contribution of abiotic ester hydrolysis to PLA degradation has been considered negligible [32]. Hence, the variation subjected by the PLA-r samples upon immersion involved morphological and contact angle changes, indicating partial surface erosion of the amorphous outer layer while ATR-FTIR analysis did not detect any degradation by-products. However, Lambert and Wagner have reported micro-particle formation of PLA immersed in demineralized water and placed in a weathering chamber at 30 °C due to the chain scission at the polymer surface [33].

In contrast, the composite dish exhibited substantial surface morphological changes due to the progressive release of inorganic particles. This process created a pitted and rough texture, increasing the material-environment interface area. Coupled with enhanced water penetration facilitated by the hydrophilic filler, this led to rapid surface modification. The comparison of the results of FTIR-ATR, TGA and DSC analyses revealed a significant and rapid decrease in filler and PBS components primarily on the sample surface and slower changes in the bulk. This resulted in alterations to the mechanical behaviour, specifically affecting the breaking parameters (tensile strength and elongation at break), while the modulus remained largely unchanged. This evolution culminated in the disintegration of the dish and the potential dispersal of fragments into the environment.

#### 3.2.3. Mb and PBAT Surface Characterization

Mater-Bi^®^ indicates a class of composite materials characterized by different compositions depending on the specific application they are intended for. The composition of shopping bags used in this research was investigated by ATR-FTIR spectroscopy, whose results are reported in Appendix A. The spectra clearly show that the Mb mainly consisted of a mixture of starch and PBAT. Although additional polyester components are usually present, no related absorptions were detected because of low concentration and overlapping bands with those of PBAT.

Appendix A shows the FTIR-ATR spectra of Mb samples immersed in freshwater for 0, 7 and 120 days. The spectra were normalized with respect to the PBAT C=O stretching absorption at 1711 cm^−1^ not superimposed to those of starch. The broad band at 3340 cm^−1^, assigned to O–H stretching of the polysaccharide, rapidly decreases after just 7 days of immersion, remaining nearly constant for longer times, as shown in Figure 5, where the normalized intensity of this band as a function of immersion time is reported.

No spectral changes were detected in the ATR-FTIR spectra of the PBAT samples (Appendix A) immersed for various durations, confirming that degradation of Mb occurs mainly at the expense of the starch component. However, as for the PLA samples, the degradation products on the sample surface cannot be detected because of their low concentration.

The starch release from Mb was clearly observed through the morphology evolution of Mb surface, as displayed in Figure 6a–c, reporting the SEM images of samples immersed for 0, 7, 60 and 120 days, respectively.

The pristine Mb sample exhibits a composite structure, with dispersed starch granules approximately 0.5 μm in diameter embedded within the PBAT matrix. Starting from the seventh day, the surface of the immersed samples shows the formation of holes due to the removal of starch granules, along with a progressive increase in roughness (Figure 6b. After 120 days, the surface of the Mb-120 sample became very uneven (Figure 6c), losing the texture characteristic of the pristine PBAT matrix or of the samples immersed for shorter durations.

The morphological evolution of the PBAT surface as a result of immersion in freshwater for 0, 60 and 120 days are displayed in Figure 6d–f. The PBAT surface, initially smooth at the sub-micrometre level, progressively roughened. After 120 days, cracks, fissures and large humps appeared, likely due to hydrolytic erosion of the outermost layer composed mainly by the amorphous phase (Figure 6f). As for the PLA-r samples, although the hydrolysis reaction was not detected by ATR-FTIR analysis, it cannot be excluded.

Despite the presence of starch, which is expected to be hydrophilic, pristine Mb showed greater hydrophobicity than PBAT, in accordance with data of the literature [34,35], which reports a water contact angle exceeding 100° for thermoplastic starch (TPS) (Figure 7).

It is attributed to the high content of crystalline amylose and the surface roughness of their samples [17,36]. Both Mb and PBAT are subjected to a progressive decrease of WCA up to a common value of 57° at 90 days of immersion. The surfaces, after the removal of starch granules, became more hydrophilic because of erosion, which increases roughness, and hydrolysis, which inserts polar –OH and –COOH functional groups, not evidenced in the ATR-FTIR analysis.

In conclusion, the comprehensive characterization revealed that immersion in freshwater induced significant morphological and compositional changes in Mb. These changes were primarily due to starch leaching, with PBAT matrix erosion becoming noticeable only after prolonged immersion. These transformations resulted in a rapid and substantial decline in mechanical properties, leading to film fragmentation.

Conversely, pure PBAT films exhibited minimal degradation or biodegradation, primarily manifested as surface morphology alterations. The potential erosion of the PBAT outermost film layer appears to predominantly affect the poly(butylene adipate)-rich fraction. However, these immersion effects resulted in only minor changes in mechanical properties, except for elongation at break, which decreased rapidly after 60 days of immersion.

#### 3.2.4. Mb and PBAT Bulk Characterization

Figure 8a shows selected thermogravimetric curves of Mb samples immersed for 0, 7 and 120 days.

The thermogram shows two weight loss regions, between 255 °C and 340 °C and between 340 °C and 440 °C. The first one is attributed to the dispersed starch and PLA trace (its presence in low fraction was inferred from DSC analysis shown later) and the second to PBAT matrix [37]. Figure 8b shows the starch content in Mb, calculated from the ratio of the first weight loss to the total weight loss between 255 °C and 440 °C, as a function of immersion time. The initial filler content of the Mb sample is approximately 30 wt%. As also evidenced by ATR-FTIR analysis, a rapid decrease in starch content occurs within the first 7 days, after which it plateaus at around 18 wt% for longer durations. In analogy with the result discussion on filler loss from PLA-d reported in Section 3.2.1, it can be presumed that the released starch originates from the outer regions of the Mb film, whereas the starch within the core, encapsulated by the PBAT matrix, is less accessible to degradation and dissolution.

The thermal stability of the PBAT copolymer remained unchanged after immersion (TGA of PBAT-0 and PBAT-120 is shown in Appendix A). This aligns with literature that reports either no change [38] or a slight decrease in decomposition temperature as a result of degradation [39,40].

The DSC thermograms of Mb and PBAT samples before and after an immersion time of 60 and 120 days are reported in Appendix A, respectively. Mb samples exhibit two melting processes in addition to a broad endothermic transition between 30 and 100 °C, attributed to residual water evaporation. The pristine PBAT matrix melts at about 107 °C, the transition shifting to higher temperatures after immersion (T_m_ = 113 °C for Mb-60 and T_m_ = 117 °C for Mb-120). This slight increase has already been observed and attributed to a crystalline structure perfectioning favoured by a decrease of molecular weight and by amorphous phase degradation [38,41]. The peak at about 160 °C has been assigned to a small fraction of PLA blended with PBAT [42]. The enthalpy of fusion, indicative of low PLA content or crystallinity, is approximately 1.5 J g^−1^ and remains relatively constant across immersion times. This suggests that PLA does not significantly degrade into the freshwater medium. Garalde et al. [43] attributed the high temperature peak to thermo-plastic starch fusion, but the constancy of the enthalpy related to this transition is in contrast with the evidence of a decrease in starch content highlighted by TGA results (Figure 4b), thus confirming the presence of PLA.

Two endothermic transitions were observed in PBAT films (Appendix A). The first one, at about 50 °C, was assigned by Zhou et al. to the annealing effect due to the storage of PBAT pellets at room temperature [44] or by Garalde et al. to the fusion of small PBAT crystal domains rich in poly(butylene adipate) [43]. After being immersed in freshwater, the melting point and intensity decrease, possibly due to a progressive erosion of the aliphatic copolymer fraction being more susceptible to the hydrolysis reaction. At higher temperatures, the second endothermic peak is related to the melting of the poly(butylene terephthalate)-rich fraction occurring at T_m_ = 120 °C for PBAT-0 and T_m_ = 118 °C for PBAT-120 (Appendix A). In both samples the melting enthalpy is about 13–14 J g^−1^, evidencing a not significant variation of aromatic copolymer fraction or sample crystallinity as result of immersion. Similarly, Wei et al. reported that the melting temperatures and crystallinity of PBAT films did not change after 10 weeks of exposure to different aquatic environments [45]. On the other hand, a significant increase in PBAT film crystallinity has been observed when significant hydrolysis occurs in polymer films subjected to degradation in soil [41,46] or in oxygen-free deionized water at 80 °C, 90 °C and 100 °C [47].

Representative stress–strain curves of Mb and PBAT samples are shown in Appendix A. The variation of their mechanical properties as a function of immersion time is displayed in Figure 8c,d. The experiments on Mb were terminated at 60 days because of the specimen fragmentation at longer immersion times. Due to the reinforcing effect of starch, Mb exhibits a higher modulus compared to PBAT, its primary matrix component, but shows a reduced toughness due to a low elongation at break. Furthermore, Mb displayed a general decline in its mechanical properties following immersion in freshwater. The rapid decrease in modulus and elongation at break can be attributed to starch leaching as well as to the rapid formation of holes, observed by SEM analysis, and to the reduction of the film density. PBAT films are more resistant to degradation during immersion in freshwater, showing only minor changes in their mechanical behaviour. The general decrease of elongation at breaks can be attributed to the formation of the flaws observed in SEM images (Figure 6f) that favour fracture triggering and propagation. This observation aligns with research by Wei et al. [40], who demonstrated a substantial loss of PBAT toughness in abiotic conditions. Specifically, they found that immersing PBAT in MilliQ water for 10 weeks resulted in a decline in both tensile strength and elongation at break. This led the authors to conclude that biodegradable polymers can release significant quantities of microplastics into aquatic systems, possibly exceeding the amounts produced by non-degradable polyethylene. However, it is important to remark that the higher thickness of the PBAT sample used in this research (120 μm) compared to that of Mb film (20 μm) could hide the effect of surface degradation of biopolymer, with the material in the bulk being more effective to define the sample mechanical properties.

PBAT-starch based Mater Bi is one of the most diffuse materials in Italy and is used to produce compostable shopping or household organic waste collection bags as certified by TÜV Austria and the Italian Composters Consortium (CIC). Therefore, it was chosen in this research because of the possibility of its accidental leakage in natural environments. As expected, the starch component degraded rapidly, as highlighted by several papers [17,48]. Starch weight loss and morphological evolution are the main degradation and biodegradation signs observed by Hu et al. during composting or sea and freshwater immersion [48].

As far as infrared spectroscopy characterization of degradation, Hu et al. have reported a clear variation of ATR-FTIR spectra of PBAT and starch-PBAT composite (5% PLA + 70% PBAT + 25% starch) immersed in lake water for 1 year, gently cleaned with distilled water and dried [29]. They observed the rise of new bands in the 1600–1675 cm^−1^ region and attributed them to the formation of ketone-like structures and carboxylic groups due to photolysis or hydrolysis reactions. On the other hand, Ruggero et al. assigned the appearance of two new peaks in the same zone to amidic groups of proteinaceous residues of non-cleaned PBAT sample. Sabatino et al. [38] have reported the results of experiments carried out on Mater Bi samples immersed in a lake for 43 days, washed with hydrogen peroxide solution (15% *v*/*v*) and dried. Small variations in TGA and DSC profile were detected after immersion, while the spectroscopic analysis was used to evidence degradation of the polymeric and polysaccharide fraction in the composite. In fact, they reported that a decrease of the intensity at 1715 cm^−1^ of C=O stretching of PBAT and in the 1150–800 cm^−1^ and 1250–900 cm^−1^ ranges, due to glycosidic bond breakage. However, the interpretation of spectral results is impaired by the absence of any spectra normalization. Similarly, Kanwal et al. [39,40] have reported that PBAT treated with isolated lipase-producing bacteria or with lipase B from Candida Antarctica brought about a not well-specified weakening of the peaks of ATR-FTIR spectra and the near disappearance of the band at 2950 cm^−1^ or 2975 cm^−1^ of –CH_2_ asymmetric stretching vibration, assigning this variation to polymer degradation. No spectrum intensity normalization has been applied in these papers. Indeed, both this and previous research [17] showed no evidence of chemical composition changes in the pure PBAT or PBAT matrix of Mb using ATR-FTIR spectroscopy. The cleaning procedure employed effectively removes plastisphere residues, preventing misinterpretations of ATR-FTIR spectra. However, it is important to note that the sample rinsing may also eliminate low molecular weight degradation products adsorbed onto the film surface, potentially underestimating polymer hydrolysis.

#### 3.2.5. PHA Surface Characterization

The ATR-FTIR spectra of pristine PHBV and of the sample immersed for 120 days, reported in Figure 9a, show no evidence of bands due to hydrolysis products, in line with the other polyester analysed.

A more detailed analysis reveals that the relative intensity of some bands changed. For example, the absorption at 1178 cm^−1^ (indicated by the arrow in Figure 9a), attributed to the C–O–C stretching of the amorphous phase [49], decreases. The intensity variation of this band, normalized with respect to the absorbance at 1378 cm^−1^, not influenced by crystallinity [50] and reported in Figure 9b, indicates a progressive crystallinity increase with the immersion time.

The rapid degradability of PHBV, a well-established characteristic of this polymer [7], was substantiated by morphological analysis of the film surface after immersion in freshwater, as shown in Figure 10e–f, where the SEM images of the surface of samples at the different investigated immersion times are reported. The pristine PHBV-0 sample, initially smooth at the micrometre level, developed roughness after 7 days of immersion. Cracks and holes appeared in one month, progressively increasing in number and size. By 120 days (PHBV-120), significant and deep erosion due to the removal of hydrolysed material was evident.

The chemical and morphological changes brought about a modification of PHBV wettability in water, as shown in Figure 11, where the measured WCA as a function of the immersion time is reported.

The decrease in water contact angle resulted from both new polar functional groups, the product of hydrolysis and increased roughness. The heterogeneous surface morphology, observed by SEM, increases the scattering of WCA data at longer immersion times.

#### 3.2.6. PHA Bulk Characterization

The thermal properties of PHBV film are reported in Appendix A, where TGA and DSC curves of PHBV-0 and PHBV-120 are displayed, respectively.

TGA curves showed a narrow decomposition range for all PHBV films, beginning at approximately 262 °C and ending between 302 °C (PHBV-0) and 307 °C (PHBV-120). The T_d_^max^ values were 293 °C for the pristine sample and 298 °C after 120 days of immersion, with intermediate samples falling within this narrow temperature window.

DSC experiments revealed only minor alterations in the thermal properties of the samples (Appendix A). All starting films exhibited a semi-crystalline structure without evidence of cold-crystallization. The typical PHBV double melting endotherms at 176 °C and 187 °C (PHBV-0) and 176 °C and 188 °C (PHBV-120) are associated with the melting of imperfect crystallites, recrystallization and re-melting. The enthalpies of fusion of PHBV-0 sample (ΔHm = 105 J g^−1^) is slightly lower than that of the film immersed in freshwater (ΔHm = 125 J g^−1^), evidencing a small increase of the crystallinity, due to plasticization activity of water, like in PLA, or to the hydrolysis and removal of amorphous fraction.

The selected stress–stress curves, reported in Appendix A, show that the PHBV is a brittle and rigid material. The variation of modulus (E), tensile strength (TS) and elongation to break (ε_b_) of PHBV as a function of immersion time is displayed in Figure 12.

This material exhibited higher modulus and lower elongation at break compared to other tested polymers and plastics. Immersion in freshwater further increased modulus and decreased elongation at break, while tensile strength remained relatively stable. The increased rigidity is likely due to increased crystallinity and the decreased elongation at break to surface flake formation. This heightened brittleness and rigidity resulted in the rupture and fragmentation of many PHBV film specimens, rendering them unusable for further testing.

Among the polyesters examined, PHBV exhibited the highest degree of degradation or biodegradation, as evidenced by significant changes in morphology and wettability. Like the other biopolymers, however, ATR-FTIR analysis did not provide direct evidence of ester bond hydrolysis. The initial PHBV films were crystalline, characterized by stiffness and, consequently, brittleness, which further increased with immersion. This leads to film fragmentation and environmental dispersion. However, in contrast to the other polymers, this is less of a concern due to PHBV rapid degradation.

#### 3.2.7. PP Characterization

The oil-based reference polymer is a 20 μm polypropylene (PP) film, the material used to wrap cigarette boxes, easily dispersible in the environment accidentally or due to bad habits. This film comprised a PP base, a thin ethylene-propylene copolymer surface layer for heat welding and an unidentified slip additive. The ATR-FTIR spectrum of the pristine PP sample (Appendix A) shows weak absorptions at 730 cm^−1^ and at 1725 cm^−1^, corresponding to the polyethylene fraction (–CH_2_ rocking) on the film surface and, likely, to an additive not removed by ethanol rinsing. The 1725 cm^−1^ absorption disappeared after 30 days, while the polyethylene contribution diminished, with only traces remaining after 120 days of immersion (Appendix A). No new bands due to oxidation phenomena were observed after the film immersion. Figure 13 displays SEM images of PP-0, PP-30, PP-60 and PP-120 samples.

The pristine film (PP-0) exhibits a morphology of twisted and intertwined lamellar texture. After the shorter immersion times, only a few holes were observed on the PP surface (Figure 13b,c). However, after 120 days (PP-120), the film became more corrugated and grooved, though the lamellar structure remained (Figure 13d). These changes suggest the partial removal of the compliant ethylene-propylene copolymer outer layer caused by mechanical abrasion due to the natural fluctuation of the samples hung in the net box.

All these light surface modifications did not change the other PP features investigated as a function of the immersion time, namely, WCA and thermal and mechanical properties.

Following prolonged immersion in freshwater, PP samples exhibited negligible degradation, with the only notable change being a modification in film morphology at 120 days. This observation contrasts with other studies that have reported alterations in FTIR spectra and surface water wettability. However, this inconsistency could be due to differences in cleaning protocols. As emphasized by Sandt et al. [51], FTIR analysis can be compromised by biocontaminants, including proteinaceous and fat residues, which can produce signals that overlap with oxidation signals. Additionally, variations in spectral analysis methods, including the interpretation of general FTIR band intensity reductions without spectra normalization or the calculation of carbonyl indices without proper spectral range consideration, may contribute to the observed discrepancies [52].

Definitive degradation markers are frequently only observed in PP samples recovered from natural environments, where the degradation history, such as exposition time or environment condition, is unknown.

### 3.3. Quantitative and Imaging-Based Analysis of Microbial Colonization on Polymers

The temporal dynamics of microbial colonization, assessed through 16S rRNA gene quantification (Appendix A), revealed distinct substrate- and time-dependent patterns, shaped by polymer composition and environmental degradability. Biodegradable polymers, including Mb, PBAT, PHBV and PLA derivatives, supported significantly higher levels of microbial colonization compared to the petrochemical-based control, PP.

PBAT exhibited a distinct colonization profile, with bacterial abundance peaking at day 60 [(1.97 ± 0.04) × 10^9^ 16S rRNA gene copies cm^−2^ (GC cm^−2^)], followed by a substantial decrease by day 120 [(1.1 ± 0.1) × 10^7^ GC cm^−2^]. The generation of hydrophilic degradation products may have promoted microbial adhesion by increasing surface wettability and offering accessible carbon sources. The subsequent decrease in bacterial abundance could be associated with surface erosion and the release of low molecular weight fragments, potentially reducing available colonization niches and nutrient sources. These findings suggest that microbial colonization on PBAT could be initially stimulated by abiotic hydrolysis, which generates favourable conditions for biofilm establishment but may subsequently decrease as readily degradable substrates are exhausted and surface integrity progressively deteriorates.

A comparable temporal trend was observed for Mb, with the highest microbial 16S rRNA gene abundance recorded at day 30 [(5.2 ± 0.6) × 10^9^ GC cm^−2^], followed by a gradual decrease at day 60 [(2.7 ± 0.6) × 10^9^ GC cm^−2^] and day 90 ((2.2 ± 0.1) × 10^9^ GC cm^−1^), and a more pronounced reduction by day 120 [(4.4 ± 0.5) × 10^7^ GC cm^−2^]. This significant early colonization, corroborated by CLSM observations (Appendix A), likely reflects the presence of a starch-based fraction within the polymer matrix, which serves as a readily bioavailable carbon source for heterotrophic bacteria. ATR-FTIR spectroscopy and TGA indicated rapid hydrolysis of the starch component within the first 7 days of immersion. SEM imaging revealed selective biodegradation or dissolution of starch granules, evidenced by surface pitting and increased roughness starting from day 7. The release of soluble degradation products may have supported initial microbial proliferation. As this labile starch fraction was progressively depleted, microbial abundance correspondingly decreased, consistent with the persistence of the more recalcitrant PBAT matrix. ATR-FTIR spectra suggested minimal chemical alteration of PBAT throughout the 120-day immersion. The comparison between ATR-FTIR and TGA analyses further supported this interpretation, indicating that starch degradation primarily occurred in the superficial layers, while starch encapsulated within the PBAT matrix remained less accessible. DSC revealed a modest increase in PBAT crystallinity over time, likely reflecting preferential erosion of amorphous domains, which may limit subsequent enzymatic hydrolysis.

PHBV showed a colonization peak at day 60 [(1.93 ± 0.02) × 10^9^ GC cm^−2^], which correlated with progressive surface degradation observed by SEM. Starting from day 7, the polymer surface developed cracks and holes, hosting microbial cells and filaments, as shown by CLSM analysis (Appendix A), which intensified over time, leading to pronounced erosion by day 120. These surface alterations increased roughness and porosity, thereby lowering the water contact angle and enhancing surface wettability. The improved wettability likely allowed microbial attachment, facilitating biofilm formation during the initial to intermediate immersion periods. The subsequent reduction in microbial abundance may be associated with the gradual increase in polymer crystallinity, as demonstrated by DSC analysis. Furthermore, mechanical testing revealed an increase in stiffness and a decrease in elongation at break over the immersion period, consistent with surface flake formation and embrittlement. This progressive fragmentation likely resulted in a partial loss of available surface area for colonization, thereby further contributing to the observed decline in bacterial abundance.

In contrast, PLA-r and PLA-d samples showed lower maximum bacterial abundance [PLA-r at day 90: (7.3 ± 0.2) × 10^8^ GC cm^−2^; PLA-d at day 60: (6.1 ± 0.4) × 10^8^ GC cm^−2^)]. This trend may be related to the intrinsic resistance of PLA to hydrolysis, likely due to its high crystallinity and hydrophobic nature. ATR-FTIR spectra showed no indications of chain scission; however, they suggested filler loss and surface pitting, which could increase surface roughness and wettability. These alterations might have facilitated microbial adhesion, although colonization appeared predominantly associated with surface imperfections and filler-rich regions rather than the PLA matrix itself. Interestingly, after approximately 90 days of immersion, a general decrease in bacterial 16S rRNA gene abundance was observed across most biopolymers. As previously discussed, this trend may be partially explained by increased polymer crystallinity and progressive surface erosion, both of which reduce enzymatic accessibility and substrate availability. However, CLSM observations (Appendix A) clearly showed the gradual colonization of polymer surfaces by photosynthetic eukaryotes, including green algae and diatoms. The production of extracellular polymeric substances (EPS) and photosynthates by these organisms may have reshaped the biofilm structure and trophic interactions. Such ecological succession could shift biofilm metabolism towards alternative carbon sources, potentially reducing the selective pressure for enzymatic degradation of the polymer substrate.

Overall, these results highlight the complex interplay between polymer physicochemical properties and biofilm development under natural freshwater conditions. While initial microbial colonization is often driven by surface accessibility and the presence of labile components, long-term persistence and degradation potential appear strongly constrained by polymer crystallinity, hydrophobicity and ecological succession within the plastisphere.

## 4. Discussion

Limited literature exists concerning the degradation analysis of biopolymers and biopolymer-based commercial products in natural freshwater ecosystems [53]. Most published studies investigate degradation processes under laboratory conditions, such as controlled anaerobic co-digestion, aerobic condition or artificial seawater or freshwater [32,42,54,55]. Additional research often focuses on the characterization of plastic debris naturally weathered in landfill, seawater, marine or lake shorelines and riparian sediments [17,20,23,25,36,42,49,50,51,52,53,54,55,56,57,58,59], where the prior sample degradation history remains unquantified. In this study, a comprehensive chemical, physical, morphological and mechanical characterization of selected materials was performed directly on samples retrieved from lacustrine freshwater at predetermined temporal intervals.

Careful sample preparation and subsequent cleaning after immersion were determined to be critical for obtaining consistent and reproducible analytical data, particularly for surface-sensitive techniques. Once these protocols were established, the comparative analysis of property evolution in pure polymers and those employed as composite matrices become feasible.

The key findings of this study are as follows:ATR-FTIR analysis revealed no direct evidence of biodegradation or degradation in pure polymers or the polymer matrices of commercial plastics, presumably due to the removal of low molecular weight degradation products during the immersion;degradation or biodegradation processes primarily occurred at the sample surface at the expense of the amorphous polymer phase;so-called compostable materials exhibited leaching of organic (starch in Mater-Bi^®^) and inorganic (calcium carbonate and talc in PLA-based dishes) fillers, with negligible or limited degradation of the polymer matrix;in PLA-based dishes, the leaching of fillers promoted the formation of voids, increasing the surface area available for subsequent degradation processes;immersion in water led to an increase in the crystallinity of pure PLA and PHBV films. This increased crystallinity, which may slow down the degradation process, embrittles the samples, thereby facilitating their fragmentation and dispersion in the environment;morphological analysis indicated that PHBV appears to degrade at a faster rate compared to the other polyesters investigated;the polypropylene (PP) film exhibited surface corrugation, likely attributable to deformation of the outermost layer composed of a compliant propylene-ethylene copolymer. However, its bulk properties remained largely unchanged;temporal variations in microbial colonization appeared closely linked to differences in polymer structure and environmental degradability.

Overall, our findings highlight the limited degradation of BPs after 120 days of immersion under natural freshwater conditions, regardless of their expected biodegradability. This suggests that laboratory tests may overestimate degradation potential and therefore require comparison with data obtained through the complexity of open environments. In situ, BPs undergo a range of modifications driven by both abiotic and biotic factors. Our results highlight the critical interplay between polymer physicochemical properties and microbial colonization dynamics. While early biofilm development is likely promoted by the availability of labile components and accessible surfaces, long-term degradation appears increasingly constrained by polymer crystallinity increase, starch concentration decrease, hydrophobicity and the ecological succession within the plastisphere. As biofilms mature into structured, multi-layered communities, microbial populations may shift their metabolic focus toward more readily available exogenous carbon sources rather than the polymer itself, ultimately limiting further degradation.

## Figures and Tables

**Figure 1 polymers-17-02236-f001:**
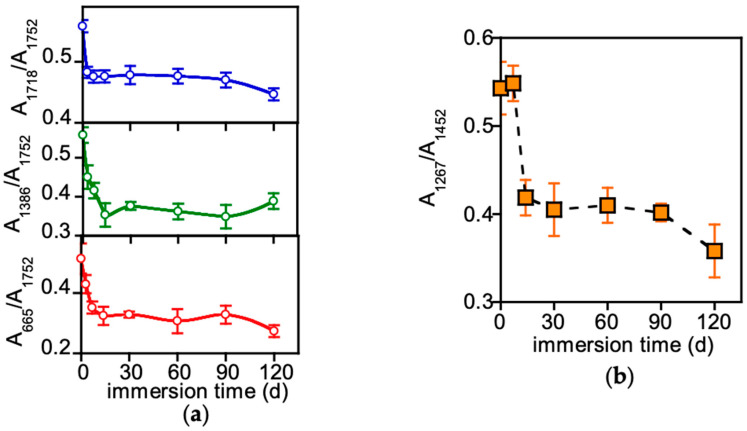
PLA-d and PLA-r ATR-FTIR characterization. (**a**) Normalized intensity variation of the bands at 1718 cm^−1^, 1386^−1^ and 665 cm^−1^ of PLA-d as a function of immersion time. (**b**) Absorbance ratio of the amorphous band at 1267 cm^−1^ and reference band at 1452 cm^−1^ of PLA-r as a function of immersion days.

**Figure 2 polymers-17-02236-f002:**
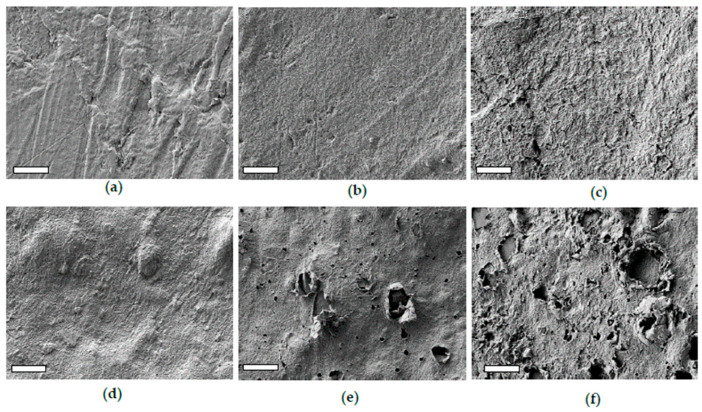
SEM images of the surface of PLA-r acquired before (**a**) and after an immersion time of 30 days (**b**) and 120 days (**c**). SEM images of PLA-d acquired before (**d**) and after an immersion time of 30 days (**e**) and 120 days (**f**). The bars on the SEM images correspond to 5 μm.

**Figure 3 polymers-17-02236-f003:**
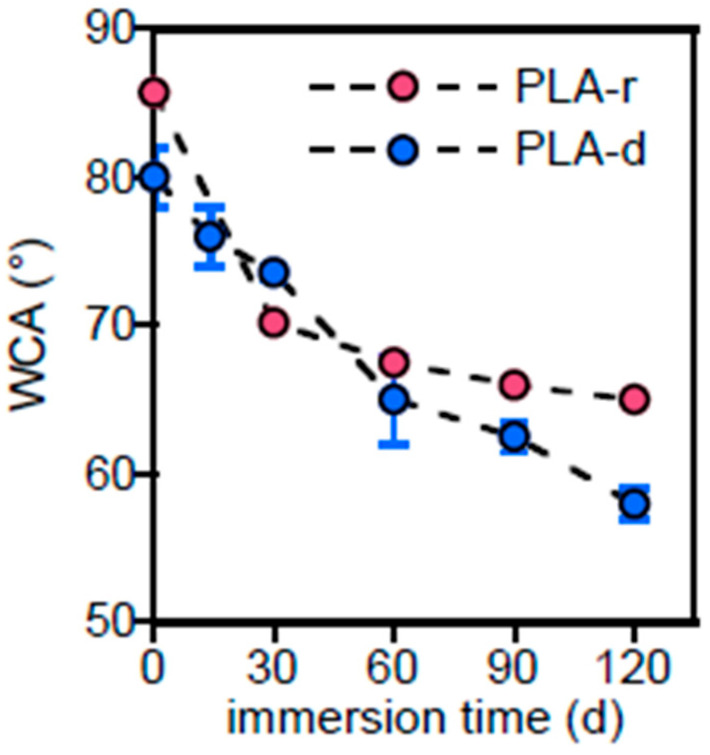
Water contact angle (WCA) variation of PLA-r and PLA-d samples as a function of immersion time.

**Figure 4 polymers-17-02236-f004:**
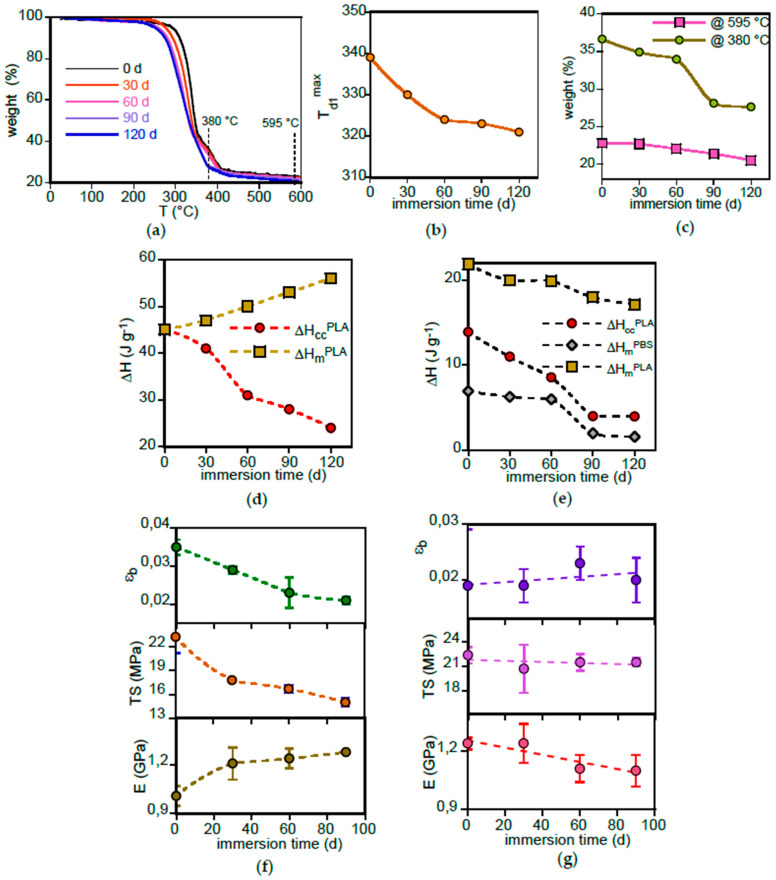
PLA-d and PLA-r bulk characterization. (**a**) TGA profile of PLA-d samples immersed in freshwater for different time periods. (**b**) Variation of the temperature at which the sample decomposes at the maximum rate (T_d1_^max^) as a function of immersion time. (**c**) PLA-d weight variation recorded at the two reference temperatures 380 °C and 595 °C as a function of immersion time. (**d**) Integrated enthalpy of cold crystallization (ΔH_cc_^PLA^) and melting (ΔH_m_^PLA^) processes of PLA-r samples as a function of immersion time. (**e**) Integrated enthalpy of cold crystallization (ΔH_cc_^PLA^) and melting (ΔH_m_^PBS^, ΔH_m_^PLA^) processes of PLA-d samples as a function of immersion time. Variation of modulus (E), tensile strength (TS) and elongation to break (ε_b_) of PLA-d (**f**) and PLA-r (**g**) as a function of immersion time.

**Figure 5 polymers-17-02236-f005:**
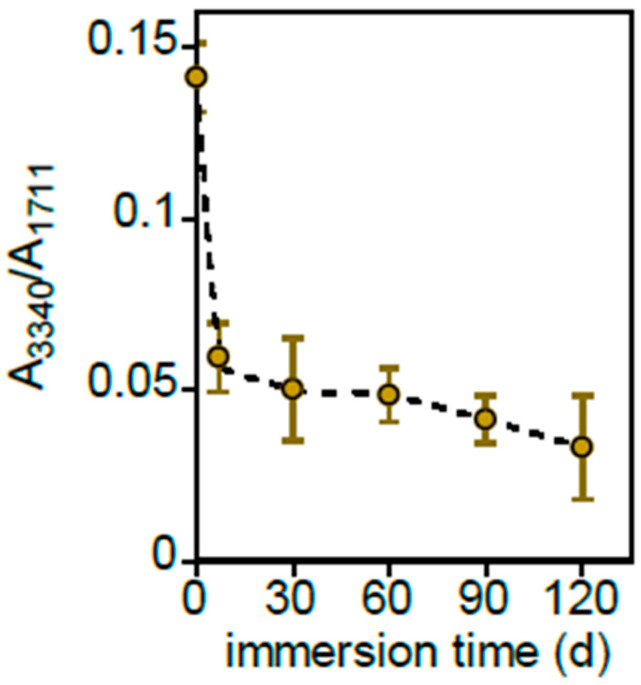
Normalized intensity variation of the band at 3340 cm^−1^ as a function of immersion time.

**Figure 6 polymers-17-02236-f006:**
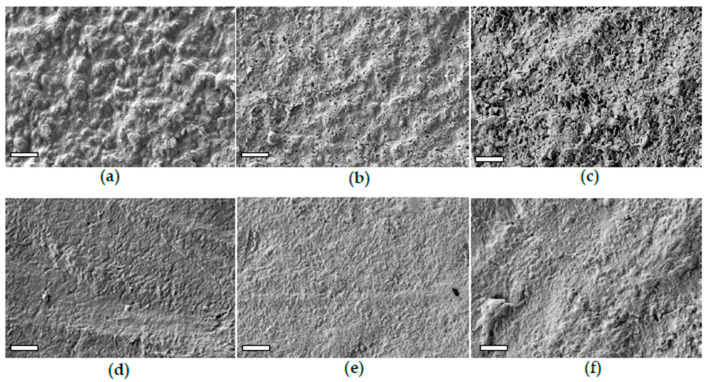
SEM images of the surface of Mb acquired before (**a**) and after an immersion time of 7 (**b**) and 120 days (**c**). SEM images of PBAT acquired before (**d**) and after an immersion time of 30 (**e**) and 120 days (**f**). The bars on the SEM images correspond to 5 μm.

**Figure 7 polymers-17-02236-f007:**
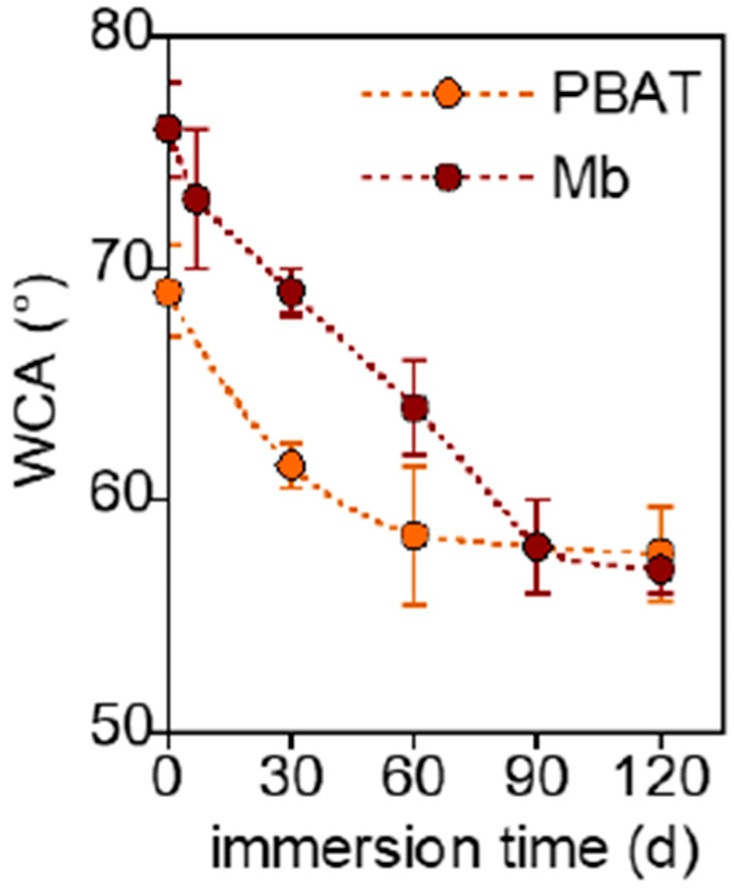
Variation of water contact angle of Mb and PBAT as a function of immersion time.

**Figure 8 polymers-17-02236-f008:**
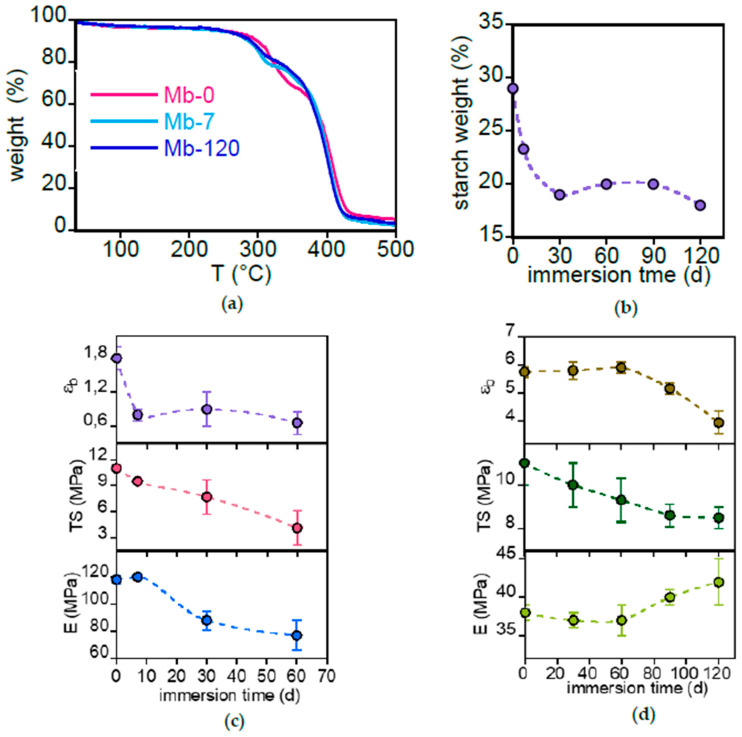
Mb and PBAT bulk characterization. (**a**) TGA curves of Mb recorded after 0, 7 and 120 days of immersion. (**b**) Starch content variation as a function of immersion time. Variation of modulus (E), tensile strength (TS) and elongation to break (εb) of Mb (**c**) and PBAT (**d**) as a function of immersion time.

**Figure 9 polymers-17-02236-f009:**
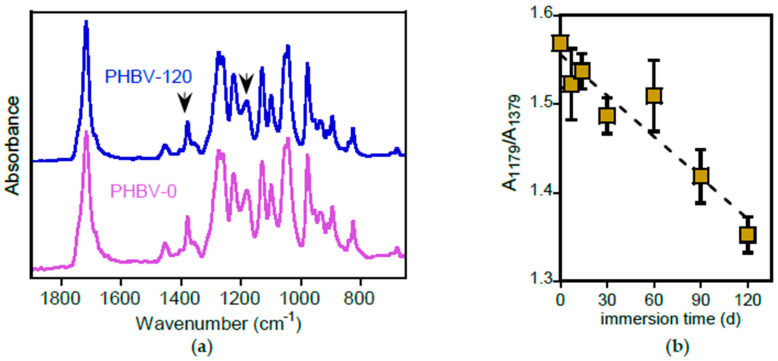
PHBV surface characterization. (**a**) ATR-FTIR spectra of PHBV-0 and PHBV-120. (**b**) Variation of the absorbance ratio A_1178_/A_1720_ as a function of immersion time.

**Figure 10 polymers-17-02236-f010:**
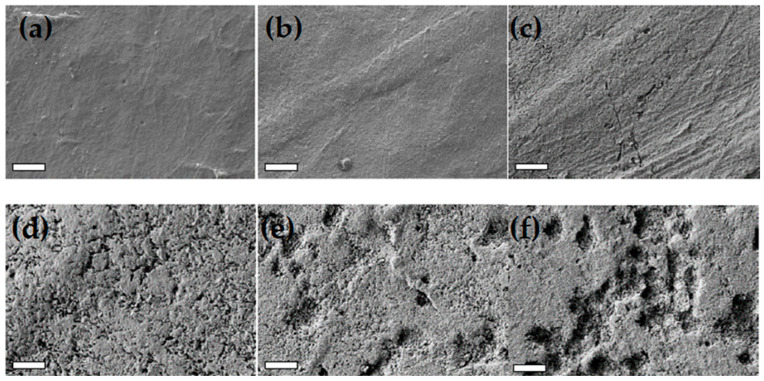
SEM images of the PHBV surface acquired after an immersion time of 0 (**a**), 7 (**b**), 30 (**c**), 60 (**d**), 90 (**e**) and 120 (**f**) days. The bars on the SEM images correspond to 5 μm.

**Figure 11 polymers-17-02236-f011:**
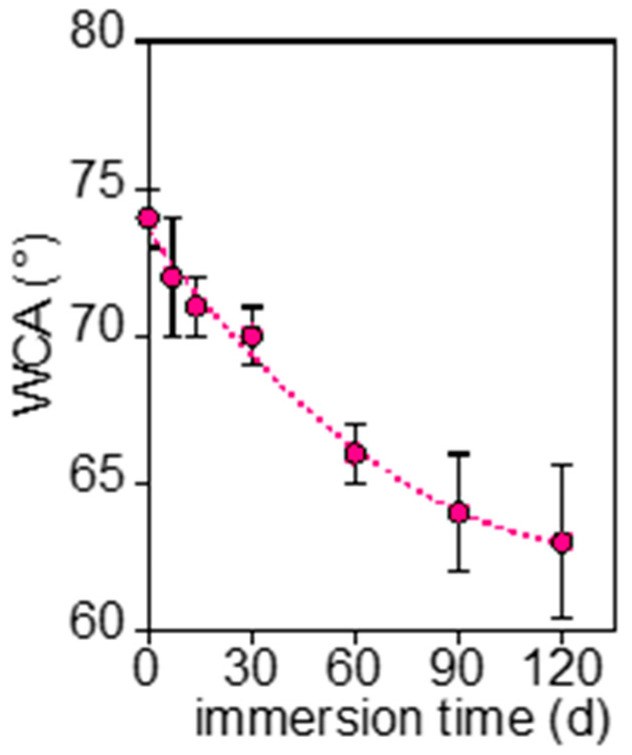
Variation of water contact angle of PHBV as a function of immersion time (d).

**Figure 12 polymers-17-02236-f012:**
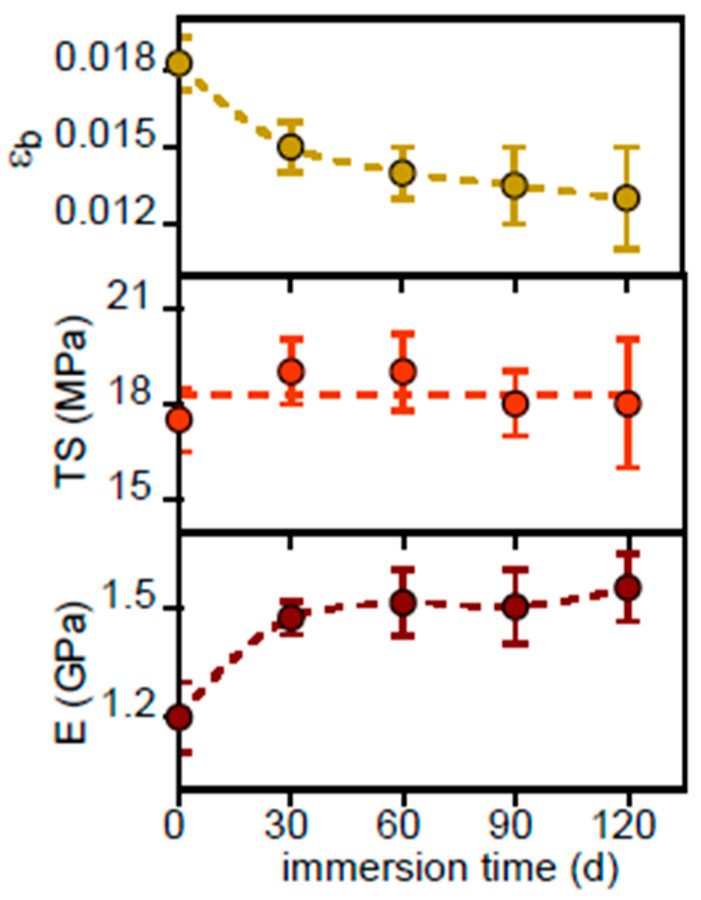
Variation of modulus (*E*), tensile strength (TS) and elongation to break (ε_b_) of PHBV as a function of immersion time.

**Figure 13 polymers-17-02236-f013:**
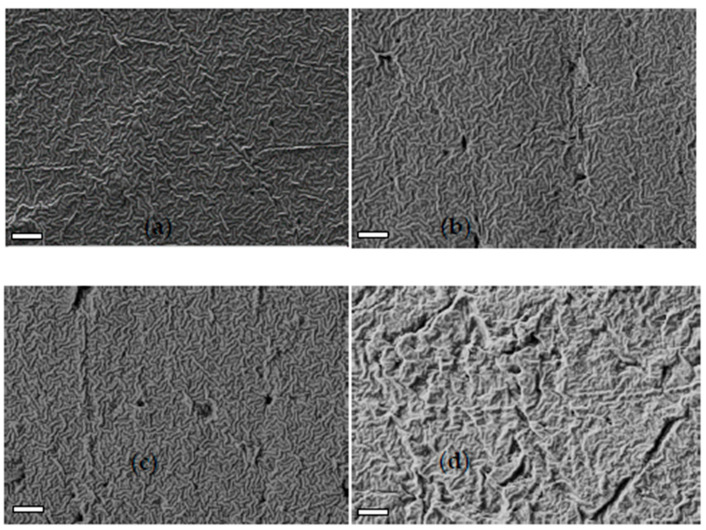
SEM characterization of PP sample immersed for 0 (**a**), 30 (**b**), 60 (**c**) and 120 (**d**) days.

**Table 1 polymers-17-02236-t001:** Sample, sample code, main features, thickness, tradename and supplier of the studied materials.

Sample ^1^	Sample Code ^2^	Features	Thickness (μm)	Tradename and Supplier
virgin PLA	PLA-r-X	- M_w_ = 55.4 kg mol^−1^ - D-Lactic acid = 1.2%	120	Ingeo 3251D NatureWorks
PLA-based dish	PLA-d-X	- compostable - composite	~200	from market
PBAT-based Mater-Bi^®^ shopper	Mb-X	- compostable - composite	20	from market
virgin PBAT	PBAT-X	- compostable - M_w_ = 74 kg mol^−1^	120	Ecoflex^®^ C1200 BASF
virgin PHBV	PHBV-X	- HV unit = 3 mol % - Mw = 590 kg mol^−1^	120	ENMAT Y1000 TianAn Biopolymer
Polypropylene	PP-X	- ethylene-propylene copolymer sealable coating	20	cigarettes secondary packaging

^1^ PLA = polylactic acid, PBAT = poly(butylene adipate-co-terephthalate), PHBV = poly(3-hydroxybutyrate-co-3-hydroxyvalerate). ^2^ X represents the immersion days.

## Data Availability

The original contributions presented in this study are included in the article/Appendix A. Further inquiries can be directed to the corresponding authors.

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
