# Peer review of "Biodegradation in Freshwater: Comparison Between Compostable Plastics and Their Biopolymer Matrices"

_polymers, 2025, doi:10.3390/polym17162236_

Round 1

Reviewer 1 Report

Comments and Suggestions for Authors

579 / 5,000

  This manuscript presents a four-month field study investigating the biodegradation of compostable plastics and their pure polymer matrices in a lentic freshwater ecosystem. The authors compare commercial products (Mater-Bi® shopping bag and PLA-based disposable plate) with their respective pure polymer matrices (PBAT and PLA), using PHBV and polypropylene as reference materials. The study employs comprehensive analytical techniques, including morphological, chemical, thermal, and mechanical characterization, as well as microbial colonization analysis.

- Abstract:

The abstract statement "Results showed limited degradation of pure polymers and their matrices in commercial products after 120 days" (Page 1, Lines 24-25) requires more precise quantification. The term "limited" is subjective and should be substantiated by specific metrics or percentage values to increase scientific rigor.

- Introduction: 

The statement "Plastics pose significant environmental threats, including physical harm to aquatic organisms, disruption of food webs and the release of toxic contaminants into water systems, thereby exacerbating the degradation of freshwater habitats [1]" (Page 2, Lines 42-44) lacks specificity regarding freshwater versus marine environments. The cited reference primarily addresses marine systems, and the authors should provide more targeted evidence for freshwater-specific impacts.

The claim "However, the degradation of these BPs under natural environmental conditions, particularly in freshwater ecosystems, remains poorly understood" (Page 2, Lines 60-63) requires stronger literature support. While the authors cite Harrison et al. (2018) [4], a more comprehensive review of existing freshwater biodegradation studies would strengthen this assertion.

The definition of biodegradation as proceeding "through multiple stages: initial enzymatic bio-fragmentation, followed by the assimilation and mineralization of resulting oligomers and monomers [7]" (Page 2, Lines 71-73) is oversimplified. The authors should acknowledge the complexity of biodegradation pathways, including the role of abiotic factors and the potential for incomplete mineralization leading to microplastic formation.

The statement "A four-month field experiment was conducted in a lentic ecosystem to provide critical insights into the degradation of BPs" (Page 2, Lines 86-88) lacks justification for the 120-day timeframe. Given that biodegradation in natural environments can be extremely slow, the authors should explain why this duration was considered sufficient and acknowledge potential limitations.

- Materials and Methods: 

Section 2.1, Page 3, Table 1: The material selection lacks clear justification for thickness variations (20-200 μm). The authors state different thicknesses for materials but do not address how this variable might influence degradation rates and comparability of results. This is a significant methodological concern that could confound the interpretation of degradation differences. Furthermore, the authors should correct the information on the thickness of the PLA-based dish (PLA-d-X) in the table. The thickness value is negative (-200).

The experimental site description "a selected coastal site of Lake Bracciano (42°06'19.0"N 12°11'23.0"E, 164 m a.s.l., Latium, Italy), a crucial drinking water reservoir in Central Italy" (Page 4, Lines 129-130) lacks essential environmental parameters. Critical missing information includes water temperature ranges, pH, dissolved oxygen levels, nutrient concentrations, and microbial community baseline data. These parameters are fundamental for understanding biodegradation processes and for reproducibility.

The deployment methodology states samples were "submerged approximately 20 cm below the water surface" (Page 4, Lines 134-135). This shallow depth may not be representative of typical plastic debris distribution in freshwater systems. The authors should justify this depth choice and discuss potential limitations regarding UV exposure and temperature fluctuations.

The cleaning protocol development is commendable, but the statement "Three different procedures to remove biofilm microorganisms and chemical residuals from sample surfaces prior to characterization were tested" (Page 4, Lines 154-164) lacks quantitative validation criteria. The authors should provide specific metrics demonstrating cleaning efficiency and potential polymer damage.

The FTIR methodology states "all reported spectra and the results obtained from their analysis were normalized by selecting adequate absorption band intensities, specific for each material" (Page 4, Lines 172-175). This normalization approach lacks standardization and could introduce bias. The authors should specify which bands were used for each material and justify these choices.

- Results and Discussion: 

Results are presented in a logical sequence with appropriate figures and tables. However, the presentation lacks statistical rigor with absent error bars and confidence intervals. The interpretation of spectroscopic data tends to be qualitative rather than quantitative, and mass balance data is notably absent. The distinction between biodegradation, leaching, and fragmentation processes is not clearly established in the results presentation.

Section 3.1, Page 8, Figure 1 and associated text: The interpretation of FTIR results requires more rigorous analysis. The statement "Since the ATR-FTIR analysis has a penetration depth on the order of one micron or less, the results of Figure 1a indicate that PLA-d samples lose fillers from the outer layer within the first 14 days of immersion" (Page 8, Lines 312-316) makes a significant assumption about the relationship between spectral changes and filler loss. The authors should provide a broader discussion, as they did not also rely on elemental analysis or mass loss measurements to support this interpretation.

The analysis of PLA-r crystallinity changes states "No new bands appeared in the spectra of PLA-r, but minor variations in the intensity and shape of certain absorption bands, not attributed to matrix compositional changes but primarily influenced by changes in polymer crystallinity" (Page 8, Lines 320-323). This interpretation dismisses potential chemical degradation too readily. The authors should consider that early-stage hydrolysis might not be detectable by FTIR, especially if degradation products are removed during the cleaning protocol.

Section 3.2.2, Page 15, Figure 4 and associated text: Analysis of Mater-Bi® starch content reveals important findings, but interpretation has limitations. The statement "The initial loading content of the Mb sample is approximately 30 wt%. As also evidenced by ATR-FTIR analysis, a rapid decrease in starch content occurs in the first 7 days, after which it stabilizes at around 18 wt% for longer periods" (Page 15, Lines 549-551) suggests significant mass loss, although the authors do not provide total mass balance data. This is a critical omission for understanding true biodegradation versus simple leaching. The explanation "It can be assumed that the released starch originates from the outer regions of the Mb film, while the starch within the core, encapsulated by the PBAT matrix, is less accessible to degradation and dissolution" (Page 15, Lines 551-554) is speculative, without supporting evidence. Cross-sectional analyses or depth profiles would be needed to validate this assumption.

- Conclusions:
The conclusions are generally consistent with the data presented, but they make claims that go beyond what the results can definitively support. The assertion that "laboratory tests may overestimate degradation potential" is not fully validated by a 120-day field study. The conclusions would be strengthened by a more conservative interpretation of the limited degradation observed and by a clearer recognition of the study's limitations.

Comments on the Quality of English Language

The manuscript is generally well-written in English, though some sections would benefit from more precise scientific language. The overall structure is appropriate, but the content organization could be improved to enhance clarity and logical flow.

Author Response

Comment 1:Abstract: The abstract statement "Results showed limited degradation of pure polymers and their matrices in commercial products after 120 days" (Page 1, Lines 24-25) requires more precise quantification. The term "limited" is subjective and should be substantiated by specific metrics or percentage values to increase scientific rigor.

Response: Considering that it is a brief summary of the article, we didn't consider it necessary to specify "how much" because there are so many polymers and we would have had to provide data for each polymer. We leave it up to the reader to verify how much and how they have degraded. However, we have revised the sentence.

Comment 2: Introduction: The statement "Plastics pose significant environmental threats, including physical harm to aquatic organisms, disruption of food webs and the release of toxic contaminants into water systems, thereby exacerbating the degradation of freshwater habitats [1]" (Page 2, Lines 42-44) lacks specificity regarding freshwater versus marine environments. The cited reference primarily addresses marine systems, and the authors should provide more targeted evidence for freshwater-specific impacts.

Response: We added freshwater-specific references to support the statement, as now indicated in the revised manuscript.

Comment 3: The claim "However, the degradation of these BPs under natural environmental conditions, particularly in freshwater ecosystems, remains poorly understood" (Page 2, Lines 60-63) requires stronger literature support. While the authors cite Harrison et al. (2018) [4], a more comprehensive review of existing freshwater biodegradation studies would strengthen this assertion.

Response: We have addressed this point by adding more specific references on bioplastic degradation in freshwater environments in the revised manuscript.

Comment 4: The definition of biodegradation as proceeding "through multiple stages: initial enzymatic bio-fragmentation, followed by the assimilation and mineralization of resulting oligomers and monomers [7]" (Page 2, Lines 71-73) is oversimplified. The authors should acknowledge the complexity of biodegradation pathways, including the role of abiotic factors and the potential for incomplete mineralization leading to microplastic formation.

Response: This is nothing more than the introduction to an article which mainly deals with the characterization of polymeric materials subjected to degradation due above all to the presence of biotic organisms in freshwater. Therefore, it seems logical to provide a generalized description of the degradation mechanism, which will be the subject of another article that will go into detail. The purpose of this article is primarily to highlight the effects of degradation on the surface of polymeric materials.

Comment 5: The statement "A four-month field experiment was conducted in a lentic ecosystem to provide critical insights into the degradation of BPs" (Page 2, Lines 86-88) lacks justification for the 120-day timeframe. Given that biodegradation in natural environments can be extremely slow, the authors should explain why this duration was considered sufficient and acknowledge potential limitations.

Response: We wanted to verify and highlight the degradation speed of some biopolymers marketed and advertised as degradable in very short times. Really, this paper focuses on the initial four months of a planned immersion study of at least one year. During this time, some samples fragmented, making certain analyses, like mechanical tests, impossible. Therefore, we added an intermediate data collection point at 120 days to assess the degradation status and compare all the results obtained by different techniques. It's important to note that the experimental design was organized to realistically simulate the fate of plastics in a natural environment. This involved exposing the samples to fluctuating freshwater and wave motion conditions and the associated stresses. This is also the reason why the planned experiments on sample weight loss were not reported, as explain in a next response.

Comment 6: Materials and Methods: Section 2.1, Page 3, Table 1: The material selection lacks clear justification for thickness variations (20-200 μm). The authors state different thicknesses for materials but do not address how this variable might influence degradation rates and comparability of results. This is a significant methodological concern that could confound the interpretation of degradation differences. Furthermore, the authors should correct the information on the thickness of the PLA-based dish (PLA-d-X) in the table. The thickness value is negative (-200).

Response: As for as the PLA-d-sample, the symbol before the thickness value is ~ (about), since the thickness of the dish is not constant in all its parts (the symbol has been enlarged in the revised version) but even commercial polymeric materials exhibit fluctuations in thickness values, probably attributable to the different mechanical stresses to which they were subjected during use. Anyway, the thickness of the Mb and PP samples is determined by their industrial production process and the thickness of the PBAT samples could not be replicated with laboratory equipment. Conversely, we were unable to produce thicker Mb films. The thickness of the PLA-r film is comparable to that of PLA-d. However, it is important to note that the degradation process, particularly in the initial stages, primarily affects the outer surface and the very first layers of the sample. The surface properties, such as those measured by ATR-FTIR, WCA, and SEM, are not influenced by sample thickness. Only bulk analyses can be affected by the sample size. The crystallization of PLA-r is promoted by water absorption, which can diffuse into the sample. However, a comparison between PLA-r and PLA-d shows that their DSC thermal properties are significantly different even in the virgin samples. Their evolution during immersion can be attributed to differences in thickness only to a minor extent. The comparison of the mechanical properties of Mb and PBAT could be influenced by the different sample thicknesses. The thicker PBAT sample might be less affected by surface degradation than the thinner Mb film. However, both the existing literature and our results show that PBAT undergoes very low degradation, resulting in minimal changes to its mechanical properties. For the sake of caution, a comment was added to the revised version regarding the changes in mechanical properties: "However, it is important to remark that the higher thickness of the PBAT sample (120 μm) compared to that of the Mb film (20 μm) could mask the effect of surface degradation of the biopolymer, as the bulk material is more effective in defining the sample mechanical properties."

Comment 7: The experimental site description "a selected coastal site of Lake Bracciano (42°06'19.0"N 12°11'23.0"E, 164 m a.s.l., Latium, Italy), a crucial drinking water reservoir in Central Italy" (Page 4, Lines 129-130) lacks essential environmental parameters. Critical missing information includes water temperature ranges, pH, dissolved oxygen levels, nutrient concentrations, and microbial community baseline data. These parameters are fundamental for understanding biodegradation processes and for reproducibility.

Response: This research is focused on the phenomenological investigation on biopolymer and plastic degradation on a selected site. Considering that the water from Lake Bracciano feeds Rome's drinking water supply system, we felt it was sufficient to conclude that it possesses absolutely optimal characteristics and is within the limits established by law. The authors are aware of the importance of environmental parameters therefore during the experiment, various chemical and physical data were collected, which we have been included in Table S2 in the supplementary materials. Moreover the microbial community baseline data have been carefully evaluated as starting point for a further paper dealing with a detailed biological analysis.

Comment 8: The deployment methodology states samples were "submerged approximately 20 cm below the water surface" (Page 4, Lines 134-135). This shallow depth may not be representative of typical plastic debris distribution in freshwater systems. The authors should justify this depth choice and discuss potential limitations regarding UV exposure and temperature fluctuations.

Response: The choice to deploy the samples at approximately 20 cm below the water surface was primarily guided by the need to ensure constant immersion in freshwater and to promote colonization by naturally occurring microorganisms, in order to evaluate their potential role in the degradation of polymeric materials. The number of samples and analytical techniques we used necessarily limited our investigation into all possible conditions to which the samples could be subjected. However, from an ecological point of view, it is well known that most "life" takes place in the first 1-2 meter of water, and therefore the 20 cm were sufficient to carry out the experiment. So much so that already in the first few days of immersion, many microorganisms were found on the surface of the samples. Regarding the presence of plastics, previous monitoring studies performed in the same lake, not yet published (Life Blue Lakes project), analyses of the water column (up to 70 m deep) have shown that most microplastics are present on the surface. Temperature variations (23 - 29°C) do not appear to have altered the bacterial community, but analyzing bacterial communities IS NOT the purpose of this objective. Anyway, the main aim of this study was not to isolate specific abiotic or biotic degradation mechanisms, but rather to assess surface changes of the materials following exposure to a realistic freshwater environment. Factors such as immersion depth, distance from the shoreline, the season of immersion and temperature fluctuations—all of which influence the degradation rate—were not explored.

Comment 9: The cleaning protocol development is commendable, but the statement "Three different procedures to remove biofilm microorganisms and chemical residuals from sample surfaces prior to characterization were tested" (Page 4, Lines 154-164) lacks quantitative validation criteria. The authors should provide specific metrics demonstrating cleaning efficiency and potential polymer damage.

Response: We agree with the reviewer's objection that the original text was not clear on this point. In fact, all samples were subjected to the specified cleaning procedure (Method 3) before being immersed in freshwater, a method consistent with our previous published work (reference 17). Our analyses, including ATR-FTIR, WCA, and SEM morphology, showed no changes when compared to the virgin samples. This experimental finding has been included in the revised manuscript at the end of subsection 3.1 for clarity as in the following: “To confirm that the cleaning process did not damage the samples, we subjected all of them to the selected cleaning procedure (Method 3) before immersion in freshwater. We then characterized the samples using ATR-FTIR, WCA, and SEM. The results showed that the surface properties were unchanged compared to the virgin samples. Consequently, we adopted this new sequential cleaning procedure for all samples.”

Comment 10: The FTIR methodology states "all reported spectra and the results obtained from their analysis were normalized by selecting adequate absorption band intensities, specific for each material" (Page 4, Lines 172-175). This normalization approach lacks standardization and could introduce bias. The authors should specify which bands were used for each material and justify these choices.

Response: This is a widely used method, and usually specific peaks of functional groups present in the polymer matrix are taken as references. For example, look at the “Carbonyl index” to determine the polyethylene degradation. Anyway, the “standardization” of the normalization bands cannot be done because of the different polymers we have investigated and the different composition of plastics and polymer matrix. Additionally, ATR-FTIR analysis had different task, being used to characterize different sample properties, such as composition and crystallinity. To address the reviewer's suggestion, we have added a clear explanation for the choice of each normalization band where it was previously missing. The text was modified as follows: section 3.2.1 “The intensities of these bands were normalized with respect to the intensity of the PLA C=O stretching band at 1749 cm-1, not superimposed to those of the filles, and plotted as a function of immersion time in Figure 2a.”

Section 3.2.2. "The spectra were normalized with respect to the PBAT C=O stretching absorption at 1711 cm-1, not superimposed to those of starch"

Moreover, at line 302, the band choice for PLA-d was already specified. At line 325, the choice for PLA-r was already discussed in the text. This band cannot be used for PLA-d sample because it is superimposed to that of CaCO3

Comment 11: Results and Discussion: Results are presented in a logical sequence with appropriate figures and tables. However, the presentation lacks statistical rigor with absent error bars and confidence intervals. The interpretation of spectroscopic data tends to be qualitative rather than quantitative, and mass balance data is notably absent. The distinction between biodegradation, leaching, and fragmentation processes is not clearly established in the results presentation.

Response: Error bars have been incorporated into all figures where applicable. While repeated DSC and TGA were performed on a limited number of samples, the resulting figures still demonstrate significant trends and are characterized by minimal data scatter. While FTIR spectroscopic data are not strictly quantitative (as rarely they are), they were used to observe a trend in the sample composition. To precisely quantify changes in the weight of sample components (fillers or blended polymer) or absolute crystallinity via FTIR as a function of immersion time, additional experimental work involving the creation of calibration curves would be necessary. Such analyses are uncommon in this type of research and fall outside the scope of our current investigation. In our experimental design, we planned to track sample weight as a function of immersion time. We prepared and carefully weighed numerous film coupons for this purpose. However, from the very beginning of the immersion period, the weight data were highly scattered and showed no clear trend. A detailed examination of the samples revealed that some were losing fragments. This fragmentation was attributed to the stress the films experienced from the shaking caused by wave motion and currents. We did not intentionally eliminate this problem, as our goal was to observe the effects of immersion on plastic and biopolymer items left freely in a natural environment. The term leaching was replaced with loss to avoid specifying the cause of the observed phenomena. To ensure neutrality regarding the underlying mechanisms, the terms degradation and biodegradation were consistently cited together. The term fragmentation was used to describe the process where a film breaks into smaller, distinct parts or pieces, which we believe is an accurate application of the term.

Comment 12: Section 3.1, Page 8, Figure 1 and associated text: The interpretation of FTIR results requires more rigorous analysis. The statement "Since the ATR-FTIR analysis has a penetration depth on the order of one micron or less, the results of Figure 1a indicate that PLA-d samples lose fillers from the outer layer within the first 14 days of immersion" (Page 8, Lines 312-316) makes a significant assumption about the relationship between spectral changes and filler loss. The authors should provide a broader discussion, as they did not also rely on elemental analysis or mass loss measurements to support this interpretation.

Response: We apologize for the error in the y-axis label of Figure 1b. It was incorrectly labelled as "WCA" and has been corrected in the revised version to show the absorbance ratio between the bands at 1267 cm⁻¹ and 1452 cm⁻¹, as stated in the caption. We acknowledge that elemental analysis or mass loss measurements do not directly provide information on filler distribution. While cross-sectional or depth-profile analyses, as suggested by the reviewer, would be ideal for a direct assessment, we did not perform these specific tests. However, we believe that the interpretation of our ATR-FTIR results offers a reliable indication. To make our argument more convincing, we have expanded the discussion by rewriting the text as follows: "The variation in the normalized absorption of the bands at 665 cm¹ and 1386 cm¹, as shown in Figure 1a, indicates that the filler content decreases up to 14 days and then remains nearly constant for longer immersion times. The fact that the filler bands were still clearly detectable throughout the entire experiment, combined with the fact that ATR-FTIR analysis has a penetration depth of approximately one micron or less, suggests that the sample loses its fillers primarily within this outer, thin region." The small amount of filler lost from the PLA-d sample was also confirmed through TGA analysis.

Comment 13: The analysis of PLA-r crystallinity changes states "No new bands appeared in the spectra of PLA-r, but minor variations in the intensity and shape of certain absorption bands, not attributed to matrix compositional changes but primarily influenced by changes in polymer crystallinity" (Page 8, Lines 320-323). This interpretation dismisses potential chemical degradation too readily. The authors should consider that early-stage hydrolysis might not be detectable by FTIR, especially if degradation products are removed during the cleaning protocol.

Response: Based on the reviewer's suggestion, the text from lines 328-332 of the original manuscript has been revised. The updated text now reads as follows: "Although possible hydrolysis reactions were considered, they were not detected by ATR-FTIR analysis. This is likely due to the low concentration of -COOH and -OH groups on the polymer surface. This low concentration is likely a result of the removal of low-molecular-weight hydrolysis products during the cleaning protocol, their solubilization in water or bio-assimilation, which prevented their accumulation on the surface."

Comment 14: Section 3.2.2, Page 15, Figure 4 and associated text: Analysis of Mater-Bi® starch content reveals important findings, but interpretation has limitations. The statement "The initial loading content of the Mb sample is approximately 30 wt%. As also evidenced by ATR-FTIR analysis, a rapid decrease in starch content occurs in the first 7 days, after which it stabilizes at around 18 wt% for longer periods" (Page 15, Lines 549-551) suggests significant mass loss, although the authors do not provide total mass balance data. This is a critical omission for understanding true biodegradation versus simple leaching. The explanation "It can be assumed that the released starch originates from the outer regions of the Mb film, while the starch within the core, encapsulated by the PBAT matrix, is less accessible to degradation and dissolution" (Page 15, Lines 551-554) is speculative, without supporting evidence. Cross-sectional analyses or depth profiles would be needed to validate this assumption.

Response: Since technical issues prevented us from conducting total mass balance, as previously reported, and cross-sectional analyses not accessible during experimentation and now, we used TGA experiments to provide an alternative indication of the bulk composition of the samples. Although TGA gives relative rather than absolute measurements, it is a reliable method for describing the degradation process. The conclusion that "the released starch originates from the outer regions of the Mb film, while the starch within the core, encapsulated by the PBAT matrix, is less accessible to degradation and dissolution" was inferred using the same reasoning applied to the filler loss observed in the PLA-d samples. The text from lines 551-554 of the original manuscript has been updated to the following: "In analogy with the discussion on filler loss from PLA-d reported in section 3.2.1, it can be presumed that the released starch originates from the outer regions of the Mb film, whereas the starch within the core, encapsulated by the PBAT matrix, is less accessible to degradation and dissolution.

Comment 15: Conclusions: The conclusions are generally consistent with the data presented, but they make claims that go beyond what the results can definitively support. The assertion that "laboratory tests may overestimate degradation potential" is not fully validated by a 120-day field study. The conclusions would be strengthened by a more conservative interpretation of the limited degradation observed and by a clearer recognition of the study's limitations.

Response: The sentence has been changed to suggest a comparison with data obtained in a natural environment:

This suggests that laboratory tests may overestimate degradation potential and therefore require comparison with data obtained through the complexity of open environments”.

Reviewer 2 Report

Comments and Suggestions for Authors

The topic of this article is extremely timely and interesting, given the need to increase the share of biopolymers in the production of packaging materials, in particular.

In order to introduce new biopolymer packaging to the market, it is essential to understand the degradation process in the environment, including freshwater. Publications on this topic often contain numerous errors in the interpretation of such research results. Therefore, it is expected that biopolymer research will be conducted carefully in interdisciplinary teams.

The authors of the reviewed article presented a well-prepared abstract. The introduction was written in an engaging manner, taking into account current research trends. References to the literature were carefully selected. This section of the article concludes with a justification of the purpose and scope of the research, as well as the selection of the materials studied.

The research presented in the article concerned seven types of materials. Their selection was well-justified. The sample preparation method raises no objections. The experimental procedure was clearly and accurately prepared. The experiment description indicates that the authors ensured that the article had sufficient samples for testing to perform at least three replicates of each study. It is worth noting the meticulous preparation of the samples for biological testing and the care taken to ensure their proper execution.

The reviewer agrees with the authors' observation that many results regarding polymer degradation are biased due to poor surface preparation of the test samples. Therefore, the authors should be commended for their decision to verify various methods of preparing the surfaces of the materials for analysis.

The research methodology was described accurately. The research results were described very carefully, drawing on the results presented in publications by other researchers.

The article is very comprehensive, but the authors presented numerous research results and a broad interpretation of these results.

One error was identified in the description of the curves in Figure S10: DSC heating curves for Mb (a). The curve description applies to two samples, and the description for Mb-120 is repeated.

Author Response

We thank Reviewer #2 for the positive and encouraging comments on our manuscript. We are particularly pleased that the reviewer appreciated the timeliness of the topic, the methodological rigor, and the care taken in both the experimental design and data interpretation.

Comments

The topic of this article is extremely timely and interesting, given the need to increase the share of biopolymers in the production of packaging materials, in particular.

In order to introduce new biopolymer packaging to the market, it is essential to understand the degradation process in the environment, including freshwater. Publications on this topic often contain numerous errors in the interpretation of such research results. Therefore, it is expected that biopolymer research will be conducted carefully in interdisciplinary teams. Response: In fact this is one of the aims of the research

The authors of the reviewed article presented a well-prepared abstract. The introduction was written in an engaging manner, taking into account current research trends. References to the literature were carefully selected. This section of the article concludes with a justification of the purpose and scope of the research, as well as the selection of the materials studied.

The research presented in the article concerned seven types of materials. Their selection was well-justified. The sample preparation method raises no objections. The experimental procedure was clearly and accurately prepared. The experiment description indicates that the authors ensured that the article had sufficient samples for testing to perform at least three replicates of each study. It is worth noting the meticulous preparation of the samples for biological testing and the care taken to ensure their proper execution.

The reviewer agrees with the authors' observation that many results regarding polymer degradation are biased due to poor surface preparation of the test samples. Therefore, the authors should be commended for their decision to verify various methods of preparing the surfaces of the materials for analysis.

The research methodology was described accurately. The research results were described very carefully, drawing on the results presented in publications by other researchers.

The article is very comprehensive, but the authors presented numerous research results and a broad interpretation of these results.

One error was identified in the description of the curves in Figure S10: DSC heating curves for Mb (a). The curve description applies to two samples, and the description for Mb-120 is repeated.

Response: The error has been corrected

Reviewer 3 Report

Comments and Suggestions for Authors

The topic of biodegradation in the natural environment is a very current and interesting issue from a researcher's perspective, especially if that environment is water. The manuscript aligns with current research trends.

1. The title of the paper is consistent with its content. The summary is well-prepared and quite comprehensive. It is worth highlighting the innovative nature of the proposed research from the outset.

2. The introduction is well-prepared and establishes the purpose of the work. As in the abstract, it is worth highlighting the novelty in the proposed approach to assessing biodegradation in an aqueous medium.

3. The cited literature is well-selected. It is current and provides a good basis for the presented research and the purpose of the work. It is quite extensive.

4. The materials used are among the most popular materials used for packaging. PP is a good choice as a reference for biodegradable plastics. It would be possible to further investigate TPS testing, but these materials degrade very quickly in high-humidity environments.

5. The adopted methodology is well described, but it's worth indicating, if possible, the standards according to which the samples, tests, or evaluations were performed – this is only a suggestion.

6. The drawings and photos are an excellent complement to the descriptions in the main text and the presentation of the results. However, it's worth separating the microscopic images. They are quite interesting and add a lot of content to the work. It would be a good idea to have the photos slightly larger and included in the work as separate drawings and numbered separately. For example, Fig. 1 should be separated.

7. The results are well described in the main text, but it might be worth presenting them in tables – such a table is missing. This would facilitate comparisons. The results are interspersed in the drawings, which would create some duplication, but the tabular presentation provided clarity.

8. The final conclusions are well developed and justified, and are expanded upon in terms of what is crucial to confirming the thesis put forward in the introduction.

Author Response

We sincerely thank Reviewer #3 for the positive and constructive feedback. We appreciate the thoughtful suggestions, which have helped us improve the clarity and overall quality of the manuscript.

The topic of biodegradation in the natural environment is a very current and interesting issue from a researcher's perspective, especially if that environment is water. The manuscript aligns with current research trends.

1) The title of the paper is consistent with its content. The summary is well-prepared and quite comprehensive. It is worth highlighting the innovative nature of the proposed research from the outset.

2) The introduction is well-prepared and establishes the purpose of the work. As in the abstract, it is worth highlighting the novelty in the proposed approach to assessing biodegradation in an aqueous medium.

3) The cited literature is well-selected. It is current and provides a good basis for the presented research and the purpose of the work. It is quite extensive.

4) The materials used are among the most popular materials used for packaging. PP is a good choice as a reference for biodegradable plastics. It would be possible to further investigate TPS testing, but these materials degrade very quickly in high-humidity environments.

5) The adopted methodology is well described, but it's worth indicating, if possible, the standards according to which the samples, tests, or evaluations were performed – this is only a suggestion.

Response: We thank the Reviewer for this valuable observation, however there are currently no standard protocols specifically designed to evaluate biodegradation in open freshwater systems. For this reason, we applied a multidisciplinary approach based on established analytical techniques to assess physicochemical and microbial changes. The methods have been detailed to ensure full reproducibility and scientific transparency. Only  the mechanical tests were performed according to ASTM standard procedures, as reported in the text.

6) The drawings and photos are an excellent complement to the descriptions in the main text and the presentation of the results. However, it's worth separating the microscopic images. They are quite interesting and add a lot of content to the work. It would be a good idea to have the photos slightly larger and included in the work as separate drawings and numbered separately. For example, Fig. 1 should be separated.

Response: All the SEM images were enlarged and putted in separate figures. Accordingly, all the figures have been renumbered

7) The results are well described in the main text, but it might be worth presenting them in tables – such a table is missing. This would facilitate comparisons. The results are interspersed in the drawings, which would create some duplication, but the tabular presentation provided clarity.

Response: To avoid redundancy and adhere to journal guidelines, we preferred that the data were presented exclusively in figures, which, in our opinion, provide a more comprehensive description of the observed phenomena. For enhanced clarity, plots illustrating the distinct behavior of plastics and related biopolymers were positioned side by side or integrated within the same figure.

8) The final conclusions are well developed and justified, and are expanded upon in terms of what is crucial to confirming the thesis put forward in the introduction.

Round 2

Reviewer 1 Report

Comments and Suggestions for Authors

The authors have provided comprehensive responses to all major concerns raised in the initial review. Most suggestions have been adequately addressed through appropriate textual modifications, addition of relevant references, and methodological clarifications. Particularly commendable improvements include: (1) the addition of freshwater-specific references to strengthen the environmental context, (2) enhanced transparency in FTIR normalization procedures, (3) validation evidence for the cleaning protocol, (4) inclusion of error bars in figures, and (5) better recognition of study limitations in the conclusions.

The authors provided reasonable technical justifications for methodological constraints that could not be fully resolved, such as thickness variations due to commercial product limitations and the inability to obtain mass balance data due to sample fragmentation under natural conditions. The explanation that this study focuses on the initial phase of a longer-term investigation provides appropriate context for the 120-day timeframe.

While some limitations persist (absence of quantitative biodegradation measurements, thickness variability effects, limited temporal scope), the authors have demonstrated good faith efforts to address reviewer concerns within the constraints of their experimental design. The implemented corrections significantly improve the manuscript's scientific rigor and clarity without compromising the validity of the core findings.